# Distinct origins and molecular mechanisms contribute to lymphatic formation during cardiac growth and regeneration

**Dana Gancz[1], Brian C Raftrey[2,3†], Gal Perlmoter[1†], Rubén Marín-Juez[4], Jonathan Semo[1], Ryota L Matsuoka[4‡], Ravi Karra[5,6], Hila Raviv[1], Noga Moshe[1], Yoseph Addadi[7], Ofra Golani[7], Kenneth D Poss[5], Kristy Red-Horse[2,3], Didier YR Stainier[4], Karina Yaniv[1]\***

[1]Department of Biological Regulation, Weizmann Institute of Science, Rehovot, Israel; [2]Department of Biology, Stanford University, Stanford, United States; [3]Institute for Stem Cell Biology and Regenerative Medicine, Stanford University School of Medicine, Stanford, United States; [4]Department of Developmental Genetics, Max Planck Institute for Heart and Lung Research, Bad Nauheim, Germany; [5]Regeneration Next, Duke University, Durham, United States; [6]Department of Medicine, Duke University School of Medicine, Durham, United States; [7]Department of Life Sciences Core Facilities, Weizmann Institute of Science, Rehovot, Israel

**\*For correspondence:**
Karina.Yaniv@weizmann.ac.il

†These authors contributed equally to this work

**Present address:** ‡Department of Cardiovascular and Metabolic Sciences, Lerner Research Institute, Cleveland Clinic, Cleveland, United States

**Abstract** In recent years, there has been increasing interest in the role of lymphatics in organ repair and regeneration, due to their importance in immune surveillance and fluid homeostasis. Experimental approaches aimed at boosting lymphangiogenesis following myocardial infarction in mice, were shown to promote healing of the heart. Yet, the mechanisms governing cardiac lymphatic growth remain unclear. Here, we identify two distinct lymphatic populations in the hearts of zebrafish and mouse, one that forms through sprouting lymphangiogenesis, and the other by coalescence of isolated lymphatic cells. By tracing the development of each subset, we reveal diverse cellular origins and differential response to signaling cues. Finally, we show that lymphatic vessels are required for cardiac regeneration in zebrafish as mutants lacking lymphatics display severely impaired regeneration capabilities. Overall, our results provide novel insight into the mechanisms underlying lymphatic formation during development and regeneration, opening new avenues for interventions targeting specific lymphatic populations.

## Introduction

The embryonic origins of lymphatic vessels have been debated for over a century, with a model claiming a venous origin for the lymphatic endothelium being predominant (*Semo et al., 2016*). However, recent work in zebrafish and mouse has uncovered additional surprising sources for lymphatic endothelial cells (LECs) during embryonic development (*Nicenboim et al., 2015*; *Martinez-Corral et al., 2015*; *Stanczuk et al., 2015*; *Mahadevan et al., 2014*; *Klotz et al., 2015*; *Pichol-Thievend et al., 2018*; *Stone and Stainier, 2019*; *Eng et al., 2019*). These include among others, vein resident angioblasts in the zebrafish trunk (*Nicenboim et al., 2015*), *c-kit*[+] cells in the mouse mesentery (*Stanczuk et al., 2015*), *VavCre*-derived yolk sac hemogenic endothelium (*Klotz et al., 2015*), and the local capillary plexus in the skin (*Pichol-Thievend et al., 2018*). Nevertheless, there is still a gap in our knowledge regarding the cellular mechanisms by which these initial lymphatic structures give rise to unique organotypic networks.

As most organs in the body, the heart contains a dense network of lymphatic vessels (*Klotz et al., 2015*; *Johnson and Blake, 1966*). Normal cardiac function relies on the cardiac lymphatic system for controlling myocardial fluid homeostasis, lipid transport, and the immune response (*Brakenhielm and Alitalo, 2019*). While the existence of cardiac lymphatics was reported already in the 17th century (*Rudbeck, 1653*), only recently have data emerged describing their development and cellular origins in mammals (*Klotz et al., 2015*; *Stone and Stainier, 2019*; *Flaht-Zabost et al., 2014*; *Norman and Riley, 2016*). In the mouse for instance, LECs were shown to enter the heart on the ventral side along the outflow tract (OFT) at approximately embryonic day 12.5 (E12.5), and later emerge on the dorsal side near the sinus venosus at E14.5. This is in response to VEGF-C expressed in the OFT mesenchyme and cardiac epicardium (*Klotz et al., 2015*; *Chen et al., 2014a*). From E14.5, LYVE1/PROX1/VEGFR3-expressing lymphatics sprout at the surface of the ventricle expanding from the base toward the apex, and reaching full maturity at postnatal day 15 (P15) (*Flaht-Zabost et al., 2014*; *Angeli and Harvey, 2015*). While lineage-tracing studies have uncovered different cell types giving rise to cardiac lymphatics (*Klotz et al., 2015*; *Stone and Stainier, 2019*; *Maruyama et al., 2019*), the putative link between cellular origins and functional properties, as well as the specific contribution of different LECs to neo-lymphangiogenesis under pathological conditions, remain largely unknown.

Acute myocardial infarction (MI) is one of the most common cardiac pathologies and a leading cause of death worldwide. Following MI in mammals, massive loss of cardiomyocytes (CMs), combined with limited regenerative capacity of the myocardium result in the formation of a collagen-based scar leading to reduced cardiac function. In recent years, there has been growing interest in the role of the lymphatic system in this process, as pro-lymphangiogenic treatments in mice have been shown to promote healing of the heart after MI by reducing fluid retention and improving inflammatory cell clearance (*Klotz et al., 2015*; *Vuorio et al., 2017*; *Henri et al., 2016*; *Ishikawa et al., 2007*; *Vieira et al., 2018*).

Zebrafish, unlike mammals, exhibit a remarkable capacity to regenerate their hearts (*González-Rosa et al., 2017*), making it an ideal model to study the origins and functions of the lymphatic endothelium following cardiac injury. With this in mind, we set out to characterize the cardiac lymphatic system of the zebrafish. Our results identify two distinct lymphatic populations in the zebrafish heart: one forming through sprouting lymphangiogenesis, and a second one, established through coalescence of isolated LEC clusters. Interestingly, these cardiac lymphatics differ in their response to signaling cues. We further demonstrate the presence of similar LEC populations in the mouse heart, suggesting that the mechanisms underlying cardiac lymphatic development are evolutionarily conserved. Finally, we address the behavior of LECs following cardiac injury in zebrafish, and show that not all lymphatics respond equally to injury, and that coalescence of isolated LECs represents the main mechanism of lymphatic growth following acute cardiac damage. Notably, we demonstrate that lymphatic vessels are required for cardiac regeneration in zebrafish, as mutants devoid of lymphatics display severely impaired regeneration capabilities. Taken together our findings provide novel insights into the origins, mechanisms of formation and heterogeneity of the cardiac lymphatic vasculature during development and regeneration. A better understanding of cardiac lymphatic formation holds great promise for developing new therapeutic interventions targeting specific lymphatic subsets.

## Results

### Morphological and molecular heterogeneity of zebrafish cardiac lymphatics

In contrast to the zebrafish systemic lymphatic vasculature, which develops within the first days of embryonic development (*Nicenboim et al., 2015*; *Yaniv et al., 2006*), we find that cardiac lymphatics start reaching the heart only after the larva-to-juvenile transition (*Parichy et al., 2009*). The first lymphatic vessels, labeled by the *TgBAC(prox1a:KalTA4,UAS:uncTagRFP)* (hereafter termed '*prox1a*') and *Tg(lyve1b:dsRed2)* (hereafter termed '*lyve1b*') transgenes, are detected in the OFT at ~21–34 days post-fertilization (dpf) (fish size 9–12 mm) (*Figure 1a–d*). These large collecting lymphatics expand during the following weeks (*Figure 1c,d*), until they cover the entire OFT by ~8 weeks post-fertilization (wpf) (fish size 14–24 mm) (*Figure 1b*) and reach their mature form, which

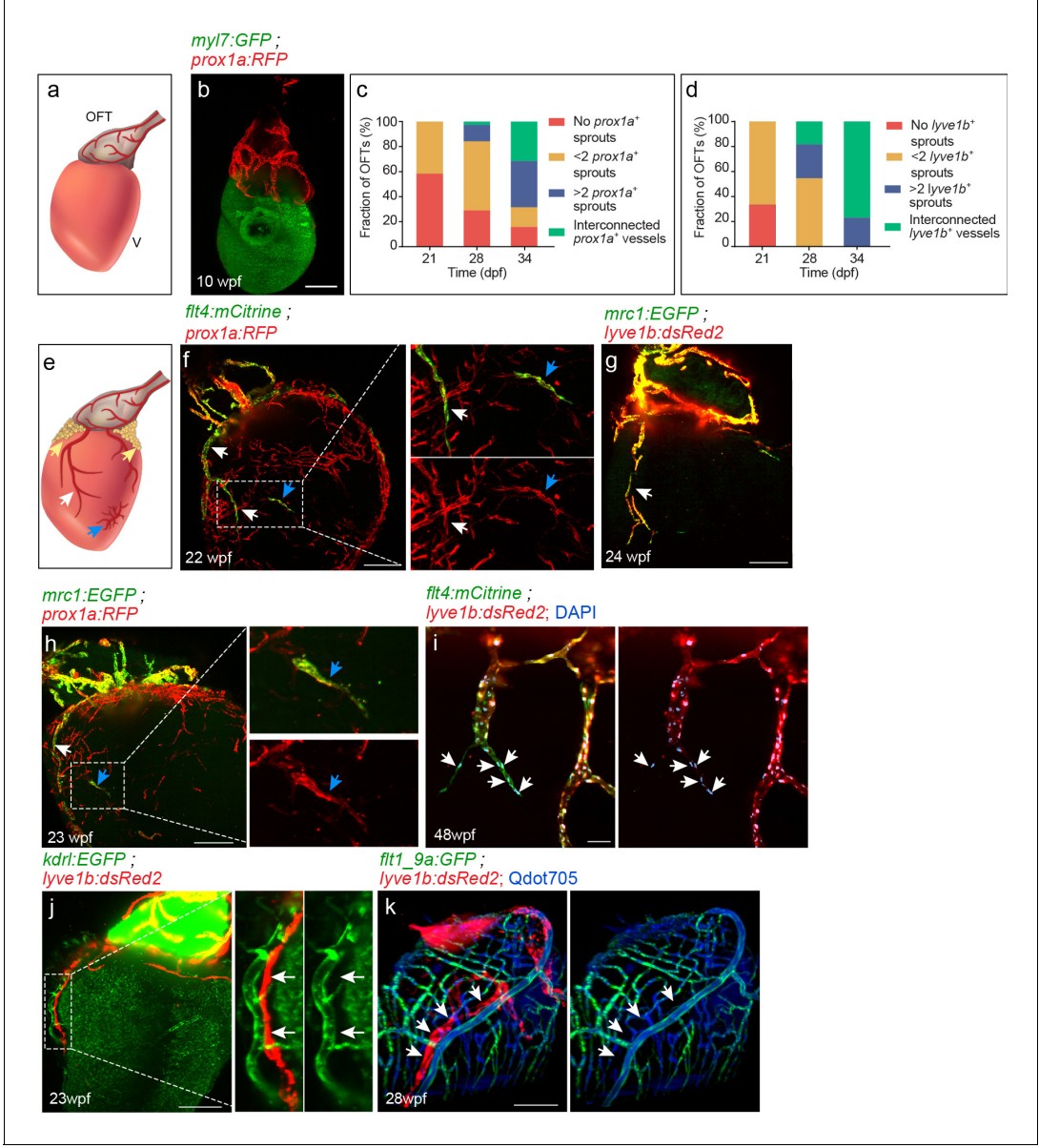

**Figure 1.** Lymphatic vessel heterogeneity in the zebrafish heart. (**a**) Diagram of ~7 wpf zebrafish heart depicting the outflow tract (OFT), ventricle (V) and lymphatics (red). (**b**) At 10 wpf (fish size 16–24 mm) *prox1a*-labeled collecting lymphatics are clearly detected in the OFT, but absent on the ventricular surface (n = 9). (**c,d**) Quantification of 21–34 dpf (fish size 7–14 mm) OFT lymphatic development in *prox1a* (**c**) ($n_{21dpf}$=48, $n_{28dpf}$=38, $n_{34dpf}$=19) and *lyve1b* (**d**) ($n_{21dpf}$=12, $n_{28dpf}$=11, $n_{34dpf}$=13) transgenic zebrafish. (**e**) Diagram of adult zebrafish heart depicting ventricular lymphatics (white arrows), fat-associated lymphatics (yellow arrows) and isolated lymphatic clusters (blue arrows). OFT and ventricular lymphatics as well as isolated lymphatic clusters in 22–24 wpf (fish size 25–30 mm) are labeled by the *flt4* (**f**), *lyve1b* (**g**), *prox1a* (**f,h**) and *mrc1a* (**g,h**) ($n_f$ = 4, $n_g$ = 5, $n_h$ = 5) transgenic reporters. (**i**) 1–5 cells at the tip of ventricular lymphatics are labeled primarily by the *flt4* transgene (arrows). Nuclei are labeled by DAPI (blue) (48wpf, fish size 25–30 mm, n = 5). (**j**) Ventricular lymphatics are not labelled by the blood vessel/endocardial- marker *Tg(kdrl:nls-mCherry)* (23wpf, fish size 25–30 mm, n = 5). (**k**) Angiogram of 28 wpf (fish size 28 mm) *Tg(flt1_9a_cFos:GFP);Tg(lyve1b:dsRed2)* heart. Cardiac lymphatics (arrows) are not labeled following intravascular injection of Qdot705 (blue) (n = 6). Scale bars are 200 μm in b, f-h, j, k; 50 μm in i. Posterior view in b, anterior view in f-k. The online version of this article includes the following video, source data, and figure supplement(s) for figure 1:

**Source data 1.** Development of OFT lymphatics.
**Figure supplement 1.** OFT lymphatics are fully established by eight wpf.
**Figure supplement 2.** *Prox1a* transgene labels cardiac lymphatics as well as a subset of coronary vessels.
**Figure 1—video 1.** Individual Z-stacks through heart shown in *Figure 1h* confirm there are no connections of isolated lymphatic cluster with the main ventricular lymphatic vasculature.
https://elifesciences.org/articles/44153#fig1video1
*Figure 1 continued on next page*

*Figure 1 continued*

**Figure 1—video 2.** Individual Z-stacks through heart shown in *Figure 1f* confirm there are no connections of isolated lymphatic cluster with the main ventricular lymphatic vasculature.

https://elifesciences.org/articles/44153#fig1video2

remains stable throughout adulthood (*Figure 1—figure supplement 1a–d*). It is not until ~12–16 wpf (fish size 25–32 mm), however, that blind-ended lymphatics arising at the base of the OFT sprout towards the ventricle, in close proximity to the major coronary vessels (*Figure 1e–h*, white arrows). At similar stages, we also detected isolated clusters of LECs spread throughout the ventricle, which were not connected to the OFT or ventricular lymphatics (*Figure 1e,f,h*, blue arrows), as well as a dense lymphatic network associated with the epicardial adipose tissue (*Figure 1—figure supplement 2a*, yellow arrows). As with other lymphatic vessels in zebrafish (*Yaniv et al., 2006*; *Jung et al., 2017*), cardiac lymphatics lack open connections with the blood vasculature as confirmed by intravascular injection of Qdots 705 (*Figure 1k*, arrows).

Examination of transgenic zebrafish with labeled LECs, revealed clear expression of *lyve1b* (*Okuda et al., 2012*), *prox1a* (*van Impel et al., 2014*), *Tg(flt4BAC:mCitrine)* (hereafter termed '*flt4*') (*van Impel et al., 2014*), and *Tg(mrc1a:EGFP)* (hereafter termed '*mrc1a*') (*Jung et al., 2017*), in all cardiac lymphatic subsets (*Figure 1f–h*, *Figure 1—videos 1* and *2*). Since these markers are absent from the blood vascularized ventricles prior to the appearance of lymphatic vessels (*Harrison et al., 2015*), we concluded that they specifically highlight LECs in the adult heart. Interestingly, we noticed that 1–5 cells at the tip of ventricular lymphatic vessels were labeled primarily by the *flt4* transgene (*Figure 1i*, arrows). Finally, we could verify that the main cardiac lymphatics were not labeled by the blood vessel marker *Tg(kdrl:nls-mCherry)* (*Figure 1j*, *Figure 1—figure supplement 1e,f*) and arterial-specific marker *Tg(flt1_9a_cFos:GFP)* (*Figure 1k*), which were clearly detected in coronary arteries and arterial capillaries.

Strikingly, we also detected sporadic expression of the *prox1a* transgene in few arterioles, that were co-labeled by the arterial enhancer *Tg(flt1_9a_cFos:GFP)* and highlighted by intravascular injection of Qdot705 (*Figure 1—figure supplement 2b*, inset). In similar fashion, assessment of hearts extracted from *prox1a;flt4* double transgenic zebrafish following Qdot705 angiography revealed two vessel populations labelled by the *prox1a* transgene– one, where the expression fully overlapped with that of the *flt4* reporter and was devoid of Qdot705 labeling (*Figure 1—figure supplement 2c*, white arrows), and a second one labelled only by *prox1a* and Qdot705 (*Figure 1—figure supplement 2c*, yellow arrows). In order to investigate whether both populations indeed express Prox1a, we carried out immunostaining with anti-Prox1 antibody. As seen in *Figure 1—figure supplement 2d* only *prox1a* positive LECs, but not *prox1a*-positive blood ECs were labelled by the Prox1 antibody (*Figure 1—figure supplement 2e*, insets), suggesting that the expression in blood ECs could be a result of post-transcriptional regulation or may represent an artefact of the transgenic reporter.

## Development of OFT lymphatics

Live imaging of the different LEC-transgenic reporters revealed that the first lymphatic sprouts reach the OFT at 21–28 dpf (*Figure 2a,b*, arrows). At this stage, no *Tg(flt1_9a_cFos:GFP)* or *Tg(fli1:EGFP)*-labeled blood vessels are detected in the heart (*Figure 2a,b*). Surprisingly, we found that these sprouts originated in four facial *lyve1b*+ vessels that run parallel to the ventral aorta (VA) (*Figure 2c,d*, yellow arrows) (*Isogai et al., 2001*) before reaching both sides of the OFT (*Figure 2d*; white arrows). These vessels, which we named 'ventral facial lymphatic (VFLs)', are also labeled by the *prox1a* transgene (*Figure 2e*, arrow) and connect to the lymphatics of the branchial arches (*Figure 2c* (LAA); e, inset). The VFLs were devoid of blood flow, as reflected by the absence of *Tg(gata1a:dsRed2)* labelled erythrocytes (*Figure 2f*, arrows), which were readily detected in the surrounding aorta and gills.

In zebrafish, the first facial lymphatic sprouts (FLS) were shown to arise in the common cardinal vein (CCV) and primary head sinus (PHS) (*Eng et al., 2019*; *Okuda et al., 2012*). Subsequently, a population of lymphangioblasts of unknown origins joins the sprouts emanating from the FLS to form individual facial lymphatic vessels. These angioblast cells, which form close to the ventral aorta

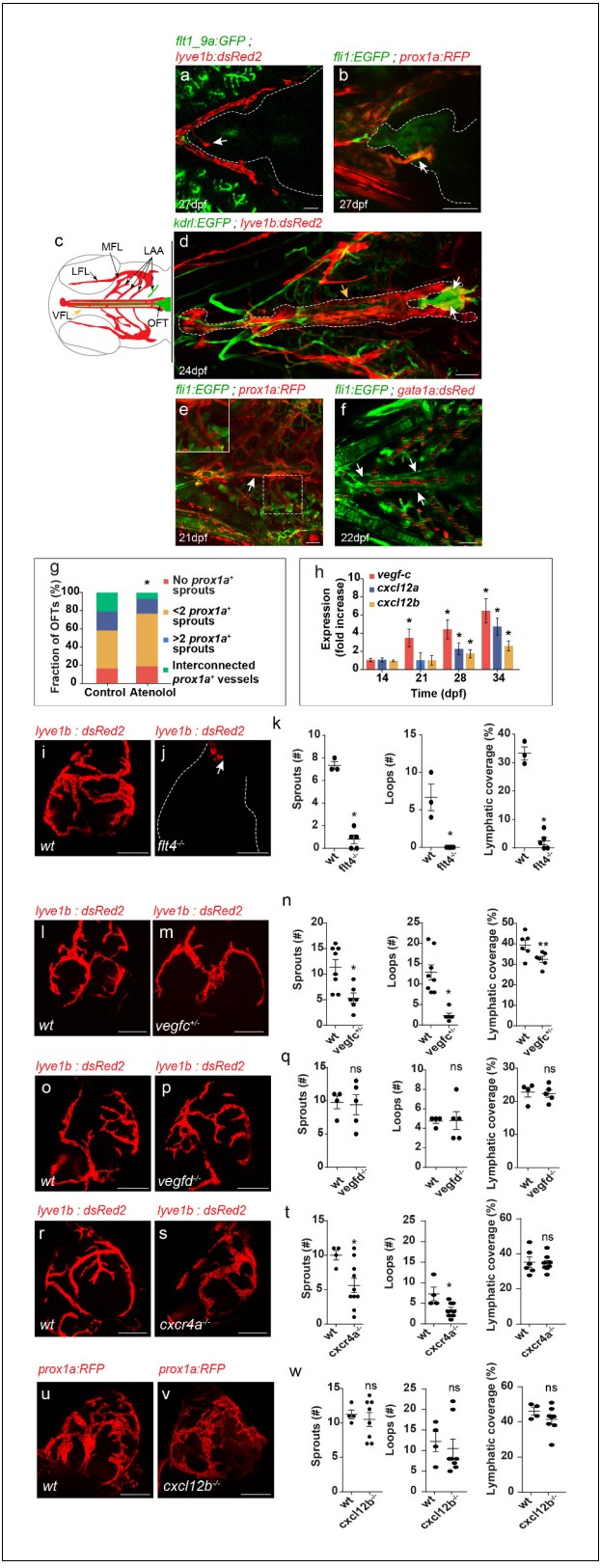

**Figure 2.** Establishment of OFT lymphatics. (**a–b**) Blood vessels are not detected in the OFT (outlined) of 21–28 dpf (fish size 5–7 mm) (**a**) *Tg(flt1_9a_cFos:GFP); Tg(lyve1b:dsRed2)* hearts (n = 6) or (**b**) *Tg(fli1:EGFP);Tg(prox1a: KalTA4-UAS:uncTagRFP)* (n = 4) fish, prior to OFT lymphatic sprouting (white arrows). (**c**) Diagram depicting a ventral view of a zebrafish larval head, indicating the approximate region imaged in (**d-f**). Facial lymphatics are *Figure 2 continued on next page*

*Figure 2 continued*

colored in red (adapted from **Okuda et al., 2012**), yellow arrow points to the VFL. (**d**) OFT lymphatic sprouts (white arrows) arising from the VFL (outlined, yellow arrow) are detected at 24 dpf (Fish size 5–7 mm) in *Tg(kdrl: EGFP);Tg(lyve1b:dsRed2);casper* larvae (n = 5). (**e**) The VFL (arrow) connects to the LAA (inset). (**f**) No blood flow is detected in the VFL (arrows) of 22 dpf (fish size 5—7 mm) *Tg(fli:EGFP);Tg(gata1a:dsRed2)* larvae (n = 10). (**g**) Quantification of OFT lymphatics in 35 dpf (fish size 9–13 mm) *prox1a* transgenic zebrafish treated with 100 μM Atenolol ($n_{control}$ = 43, $n_{Atenolol}$ = 43, *p<0.001). (**h**) mRNA levels of *vegfc*, *cxcl12a* and *cxcl12b* ($n_{independent\ experiments}$=5, *p<0.01) in the OFTs of 14–34 dpf larvae. (**i–k**) OFT of 19 wpf (22–25 mm) wt sibling (**i**) and *flt4*$^{-/-}$ (**j**) in the background of *lyve1b* demonstrating severe lymphatic defects in *flt4*$^{-/-}$ hearts, quantified in (**k**) ($n_{wt}$ = 4, $n_{flt4}$$^{-/-}$=5, *p<0.001). (**l–n**) OFT of 15 wpf (20–21 mm) wt sibling (**l**) and *vegfc*$^{+/-}$ (**m**) in the background of *lyve1b* showing malformed lymphatics in *vegfc*$^{+/-}$, quantified in (**n**) ($n_{wt}$ = 8, $n_{vegfc+/-}$=6, *p<0.01, **p<0.05). (**o–q**) OFT of 12 wpf (19–21 mm) age-matched wt control (**o**) and *vegfd*$^{-/-}$ (**p**) in the background of *lyve1b* showing normal OFT lymphatics in *vegfd*$^{-/-}$ hearts, quantified in (**q**) ($n_{wt}$ = 4, $n_{vegfd}$$^{-/-}$=5) (**r–t**) OFT lymphatics of 9.5 wpf (20–23 mm) wt sibling (**r**) and *cxcr4a*$^{-/-}$ (**s**) in the background of *lyve1b* showing mild defects in *cxcr4a*$^{-/-}$ OFT lymphatics, quantified in (**t**) ($n_{wt}$ = 4, $n_{cxcr4a}$$^{-/-}$=10, *p<0.01). (**u–w**) OFT lymphatics of 20 wpf (19–24 mm) wt sibling (**u**) and *cxcl12b*$^{-/-}$ (**v**) in the background of *prox1a* showing normal OFT lymphatics in *cxcl12b*$^{-/-}$ hearts, quantified in (**w**) ($n_{wt}$ = 4, $n_{cxcl12b}$$^{-/-}$=8). VFL, ventral facial lymphatics; LFL, lateral facial lymphatic; LAA, lymphatic branchial arches; MFL, medial facial lymphatic. Scale bars are 50 μm in a-f, 200 μm in i-v. Error bars, mean ± S.E.M.

The online version of this article includes the following source data and figure supplement(s) for figure 2:

**Source data 1.** Quantification of OFT lymphatics following Atenolol treatment.
**Source data 2.** mRNA levels of pro-lymphangiogenic cues in the OFTs of 14–34 dpf larvae.
**Source data 3.** Development of OFT lymphatics in *flt4*$^{-/-}$, *vegfc*$^{+/-}$, *vegfd*$^{-/-}$, *cxcr4a*$^{-/-}$ and *cxcl12b*$^{-/-}$ hearts.
**Figure supplement 1.** Hemodynamic changes during larva to juvenile transition modulate OFT lymphatic development.
**Figure supplement 1—source data 1.** Quantification of larvae heartbeat during development and following Atenolol treatment.
**Figure supplement 2.** VFL develops normally in lymphatic-related mutants.
**Figure supplement 3.** Generation of zebrafish *vegfd*$^{bns257}$ mutant allele.
**Figure supplement 3—source data 1.** Quantification of facial lymphatics development in *vegfd*$^{-/-}$ embryos.

and are initially devoid of both venous and lymphatic markers, contribute not only to facial lymphatics but also to the hypobranchial artery, confirming their multipotent ability (**Eng et al., 2019**). Thus, facial lymphatics, most probably including the VFLs from which the OFT lymphatics sprout, originate from local sources within the facial domain.

The appearance of the first lymphatic sprouts in the OFT correlates with the transition from the larval to the juvenile stages (**Parichy et al., 2009**). We therefore wondered whether these two processes could be somehow associated. In particular, we hypothesized that increased heart rate and cardiac output during the larval to the juvenile transition, could result in elevated blood pressure and fluid extravasation, and potentially trigger the growth and or/attraction of lymphatic vessels (**Boardman and Swartz, 2003**). To test this hypothesis, we measured the heart rate of zebrafish between 5–40 days post-fertilization (dpf), and found that it indeed peaks during the larva to juvenile transition (**Figure 2—figure supplement 1a**). Moreover, heart-rate attenuation following administration of the ß-blocker Atenolol (**Figure 2—figure supplement 1b**) (**Hein et al., 2015**) resulted in delayed appearance, and impaired branching of OFT lymphatics (**Figure 2g**, **Figure 2—figure supplement 1c,d**). To identify molecular candidates mediating lymphatic recruitment, we analyzed gene expression on isolated OFTs at the relevant stages, and detected clear upregulation of the well-established pro-lymphangiogenic factor *vegfc*, and of the chemokines *cxcl12a*, and *cxcl12b* (**Figure 2h**). In order to investigate the potential role of these factors in OFT lymphatic development, we analyzed genetic mutants. We observed marked defects in OFT lymphatics following depletion of the Vegfc receptor, Vegfr3/Flt4 (**Figure 2i–k**). *flt4* mutants bear a truncated form of the Flt4 receptor and were shown to lack a thoracic duct at 5dpf (**Kok et al., 2015**). While in general facial lymphatics were slightly shorter in *flt4*$^{-/-}$ hearts (**Figure 2—figure supplement 2a,b**, yellow arrows), no major defects were observed in the VFL, from which OFT lymphatics sprout (**Figure 2—figure supplement 2a,b**, white arrows). In contrast, lymphatic vessels were almost completely absent from the OFTs of *flt4*$^{-/-}$ hearts (**Figure 2i–k**, white arrow). Flt4 is the receptor for the pro-lymphangiogenic growth factors Vegfc and Vegfd. We therefore examined the contribution of each ligand to OFT

lymphatic development by assessing *vegfc*$^{+/-}$ (*Villefranc et al., 2013*) and *vegfd*$^{-/-}$ hearts. Since *vegfc* homozygous mutants are embryonic lethal, we examined heterozygous animals that, similar to their homozygous siblings, display reduced formation of the thoracic duct at five dpf (*Villefranc et al., 2013*), but survive through adulthood. In spite of bearing a wild type (wt) *vegfc* allele, loss of one copy of this gene resulted in maldeveloped lymphatic vessels and overall reduction of OFT lymphatic coverage in *vegfc*$^{+/-}$ fish (*Figure 2l–n*). *vegfd*$^{-/-}$ hearts (*Figure 2—figure supplement 3d,f*) in contrast, did not display any substantial defects in OFT lymphatics (*Figure 2o–q*) or in the VFL (*Figure 2—figure supplement 2e*, white arrow) at three wpf, despite of displaying minor defects during early facial lymphatic development (*Figure 2—figure supplement 3a–c*) (*Bower et al., 2017a*).

We then investigated the potential role of the Cxcr4/Cxcl12 axis in OFT lymphatic development. CXC chemokines play a well-established role in guidance of various cell types, including LECs (*Cha et al., 2012*). In addition, *cxcl12a* and *cxcl12b* were found to be expressed on the surface of the OFT at ~4 wpf, prior to the formation of OFT blood vessels (*Harrison et al., 2015*). Therefore, Cxcr44/Cxcl12 signaling can potentially mediate LEC sprouting over the OFT. Analysis of adult *cxcr4a*$^{-/-}$ hearts (*Siekmann et al., 2009*) carrying the *lyve1b* transgene revealed only minor defects in OFT lymphatics as compared to wt siblings (*Figure 2r–t*). Moreover, homozygous mutants for the *cxcl12b* ligand (*cxcl12b*$^{-/-}$) did not display any noticeable phenotypes (*Figure 2u–w*). Of note, the VFL was not affected in either *cxcr4a*$^{-/-}$ or *cxcl12b*$^{-/-}$ fish (*Figure 2—figure supplement 2f–h*).

Altogether, our results suggest that changes in heart rate during larva to juvenile transition modulate OFT lymphatic development in a Vegfc/Flt4-dependent manner, whereas Cxcr4a/Cxcl12b involvement is restricted to remodeling of the OFT lymphatic plexus.

## Establishment of ventricular lymphatics

Despite the fact that OFT lymphatics are established at 3–4 wpf, it is not until ~2 months later that they begin sprouting toward the ventricle (*Figure 1e–h*; 3a, inset). Preceding lymphatic sprouting, the zebrafish myocardium undergoes significant expansion, which was proposed to trigger the formation of the coronary vasculature (*Harrison et al., 2015*; *Gupta et al., 2013*). Interestingly, we found that ventricular lymphatics sprout in close proximity to the major coronary vessels and continue to grow and branch over the following months (*Figure 3b*). Similar association between developing lymphatics and the blood vasculature has been described in the mouse heart (*Klotz et al., 2015*). To test whether coronary vessels play an active role in ventricular lymphatic development, we treated juvenile zebrafish with Phenylhydrazine hydrochloride (PHZ), which induces CM hypertrophy and enhanced vascularization (*Sun et al., 2009*). PHZ treatment starting at ~8 wpf- coinciding with the initial development of the coronary plexus-, resulted in enlarged hearts accompanied by enhanced growth and remodeling of coronary vessels (*Figure 3c,d*). Interestingly, we also detected significantly longer lymphatic vessels in the ventricles of these animals, which closely followed major coronary vessels (*Figure 3c,d*), suggesting a possible role for the coronary vasculature in lymphatic vessel growth. To further confirm these results, we induced hypervascularization by conditionally over-expressing Vegfaa in CMs using *Tg(cmlc2:CreER);(βactin2:loxP-mTagBFP-STOP-loxP-vegfaa)* fish (hereafter termed Vegfaa-OE) (*Karra et al., 2018*) (*Figure 3e–g*, insets; *Figure 3—figure supplement 1a*). Vegfaa-OE was induced at ~7 wpf by administration of 4-hydroxytamoxifen and hearts were analyzed 17 days later. As previously reported (*Karra et al., 2018*), we detected a significant increase in blood vessel coverage, especially those of small caliber (*Figure 3e–g*, insets; *Figure 3—figure supplement 1b*), which was accompanied by a massive expansion of the lymphatic plexus (*Figure 3e–g*), supporting a tight association between blood and lymphatic vessel growth.

In order to understand whether coronary vessels are required for proper ventricular lymphatic sprouting, we first examined hearts of *cxcr4a*$^{-/-}$ fish, which have previously been reported to fail to develop coronary vasculature (*Harrison et al., 2015*). Surprisingly, we found that neither blood vessel coverage (*Figure 3h–j*), nor ventricle size (*Figure 3—figure supplement 1c*) was affected in *cxcr4a*$^{-/-}$ hearts. Instead, the coronary plexus appeared immature, lacked its stereotypic hierarchical pattern and the main arteries that run over the OFT and connect to the gills were severely malformed or absent (*Figure 3—figure supplement 1d*, yellow arrow; e, asterisk). These defects impede the establishment of proper connections with the circulatory network and rendering the cardiac muscle only partially perfused (*Figure 3—figure supplement 1f*, yellow arrows). Similar phenotypes have been described in *Cxcl12* and *Cxcr4* mutant mice, where ECs of the peritruncal plexus

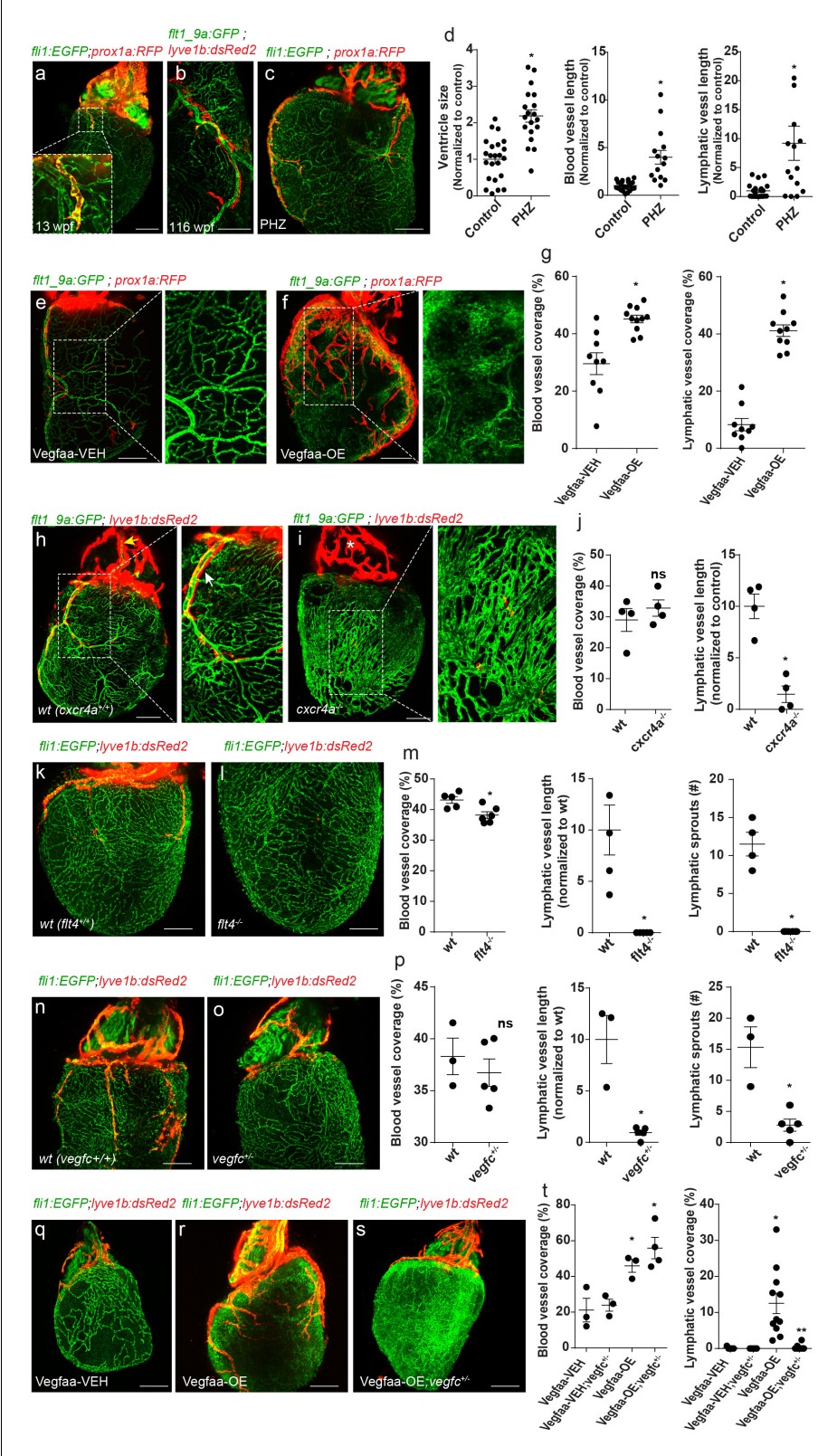

**Figure 3.** Coronary arteries serve as a scaffold for ventricular lymphatic sprouting. Insets are magnifications of dashed boxes. (a) 13 wpf (fish size 17 mm) *Tg(fli1:EGFP);Tg(prox1a:KalTA4-UAS:uncTagRFP)* hearts showing ventricular lymphatic sprouting at the base of the OFT (inset) (n = 11). (b) Ventricular lymphatics grow in close proximity to coronary arteries (inset) in 116 wpf (fish size 25–32 mm) *Tg(flt1_9a_cFos:GFP);Tg(lyve1b:dsRed2)* hearts (n = 15). (c) Heart of 11wpf (fish size 16–22 mm) *Tg(fli1:EGFP);Tg(prox1a:KalTA4-UAS:uncTagRFP)* fish treated with 100 µg/ml Phenylhydrazine

*Figure 3 continued on next page*

*Figure 3 continued*

hydrochloride (PHZ), showing increased ventricle size and total length of blood and lymphatic vessels following PHZ treatment, quantified in (d) ($n_{control}$ = 21, $n_{PHZ}$ = 15, *p<0.005). (e,f) 12 wpf (fish size 19–22 mm) *Tg(βactin2:loxP-mTagBFP-STOP-loxP-vegfaa)$^{pd262}$; Tg(cmlc2:CreER)* fish in the background of *Tg(flt1_9a_cFos:GFP);Tg(prox1a:KalTA4-UAS:uncTagRFP)*. Induction of Vegfaa-OE results in increased blood vessel (insets) and lymphatic vessel coverage, as compared to vehicle (Vegfaa-VEH) treated sibling control, quantified in (g) ($n_{Vegfaa-VEH}$ = 9, $n_{Vegfaa-OE}$ = 10, *p<0.001). (h–j) Immature coronary plexus, lacking the stereotypical tree-patterning results in nearly absent ventricular lymphatics in *cxcr4a$^{-/-}$* hearts of 22 wpf (fish size 25–28 mm) (i), as compared to wt siblings (h), quantified in (j) ($n_{wt}$ = 5, $n_{cxcr4a}$$^{-/-}$=6, *p<0.05). (k–m) Ventricular lymphatics are absent in *Tg(fli1:EGFP); Tg (lyve1b:dsRed2); flt4$^{-/-}$* hearts (l) at 19–23 wpf (fish size 25–30 mm) as compared to wt siblings (k). (m) Quantification of blood and lymphatic vessel phenotype in *flt4$^{-/-}$* hearts ($n_{wt}$ = 4, $n_{flt4}$$^{-/-}$=5 *p<0.01). (n–p) *Tg(fli1:EGFP); Tg(lyve1b:dsRed2); vegfc$^{+/-}$* hearts at 26 wpf (fish size 25–30 mm) display severely defective ventricular lymphatics. (p) Quantification of blood vessel coverage and lymphatic sprout length and number, in *vegfc$^{+/-}$* hearts ($n_{wt}$ = 3, $n_{vegfc}$$^{+/-}$=5 *p<0.005). (q–t) 12 wpf (fish size 19–22 mm) *Tg(βactin2:loxP-mTagBFP-STOP-loxP-vegfaa)$^{pd262}$; Tg(cmlc2:CreER)* fish in the background of *Tg(fli1:EGFP);Tg(lyve1b:dsRed2)*. The increase in lymphatic, but not blood vessel coverage induced by Vegfaa-OE (r), is reversed in *vegfc$^{+/-}$* heterozygous animals (s). (t) Quantification of blood and lymphatic vessel coverage in (q–s) ($n_{Vegfaa-VEH}$ = 3, $n_{Vegfaa-OE}$ = 3, $n_{Vegfaa-VEH-blood\ vessel\ coverage}$=3, $n_{Vegfaa-OE-lymphatic\ vessel\ coverage}$=10 $n_{Vegfaa-OE;vegfc\ +/-\ -lymphatic\ vessel\ coverage}$=4, *p<0.01, relative to vehicle treated sibling control, **p<0.001 relative to Vegfaa-OE). Scale bars are 200 µm. Error bars, mean ± S.E.M. All panels show anterior views.

The online version of this article includes the following source data and figure supplement(s) for figure 3:

**Source data 1.** Quantification of Phenylhydrazine hydrochloride (PHZ)-induced ventricular phenotype.
**Source data 2.** Quantification of ventricular blood and lymphatic vessel phenotype in VegfAa-OE, *flt4$^{-/-}$*, *vegfc$^{+/-}$* and *cxcr4a$^{-/-}$* hearts.
**Figure supplement 1.** Development of ventricular lymphatics.
**Figure supplement 1—source data 1.** Quantification of ventricular size in Vegfaa-OE, *flt4$^{-/-}$*, *vegfc$^{+/-}$* and *cxcr4a$^{-/-}$* hearts.

fail to properly connect to the aortic endothelium, resulting in impaired coronary artery stem formation and establishment of an immature coronary plexus (*Ivins et al., 2015*). Interestingly, the underdeveloped coronary plexus observed in *cxcr4a$^{-/-}$* hearts was associated with severe ventricular lymphatic abnormalities, characterized by markedly shorter and less branched lymphatic sprouts (*Figure 3h–j*, white arrow, *Figure 3—figure supplement 1g*), suggesting that the presence of blood ECs per se, is not sufficient to support lymphatic growth, but rather a mature coronary tree is required. To further investigate the nature of the interaction between coronary vessels and ventricular lymphatics, we assessed hearts of *flt4$^{-/-}$* and *vegfc$^{+/-}$* animals (*Figure 3k–p*). In contrast to *cxcr4a$^{-/-}$* hearts, we detected no significant differences in blood vessel coverage and patterning (*Figure 3k–p*, *Figure 3—figure supplement 1h,i*) as well as in ventricle size (*Figure 3—figure supplement 1j,k*) in these animals. Yet, ventricular lymphatics were severely affected in *vegfc$^{+/-}$* and nearly absent in *flt4$^{-/-}$* hearts (*Figure 3k–p*), indicating that Vegfc/Flt4 signaling is absolutely required for ventricular lymphatic formation. Taken together, the mutant analyses support the idea that a mature coronary plexus could serve as a source of Vegfc, a hypothesis substantiated also by the observation that mouse coronary ECs begin expressing Vegfc as they differentiate into arteries (*Su et al., 2018*). In order to test this hypothesis, we over expressed Vegfaa in *vegfc$^{+/-}$* fish. While the reduced levels of Vegfc did not impede Vegfaa-OE induced cardiomegaly and hypervascularization (*Figure 3q–t Figure 3—figure supplement 1l,m*), the increase of lymphatic coverage (*Figure 3e–g*) was abrogated in these animals (*Figure 3q–t*), supporting the idea that a mature coronary plexus serves as a scaffold for ventricular lymphatic development, in a Vegfc-dependent manner. Moreover, these results demonstrate that the increased lymphatic growth observed in Vegfaa-OE hearts results from the expanded blood vasculature and, most probably, from enhanced Vegfc production, and not from a direct effect of Vegfaa on LECs. In the future, it will be interesting to investigate whether the Cxcl12/Cxcr4 signaling pathway somehow regulates Vegfc secretion from the coronary ECs.

## A novel population of isolated LECs is detected in the hearts of zebrafish and mice

Besides the 'regular' lymphatic vessels, the zebrafish ventricle holds an additional population of isolated LECs (*Figure 1e,f,h*), labeled by the *prox1a, lyve1b, mrc1a* and *flt4* transgenic reporters (*Figure 1f,h Figure 4a-d*, *Figure 4—video 1*), and positively stained by Prox1 antibody (*Figure 4e*, arrows). Light sheet imaging of explanted hearts revealed that these lymphatic structures initially appear as single cells (~12–13 wpf, fish size 20–22 mm), that later on expand to form isolated capillaries (~16 wpf, fish size 23–28 mm) (*Figure 4b–d*, *Figure 4—video 1*), through a process that

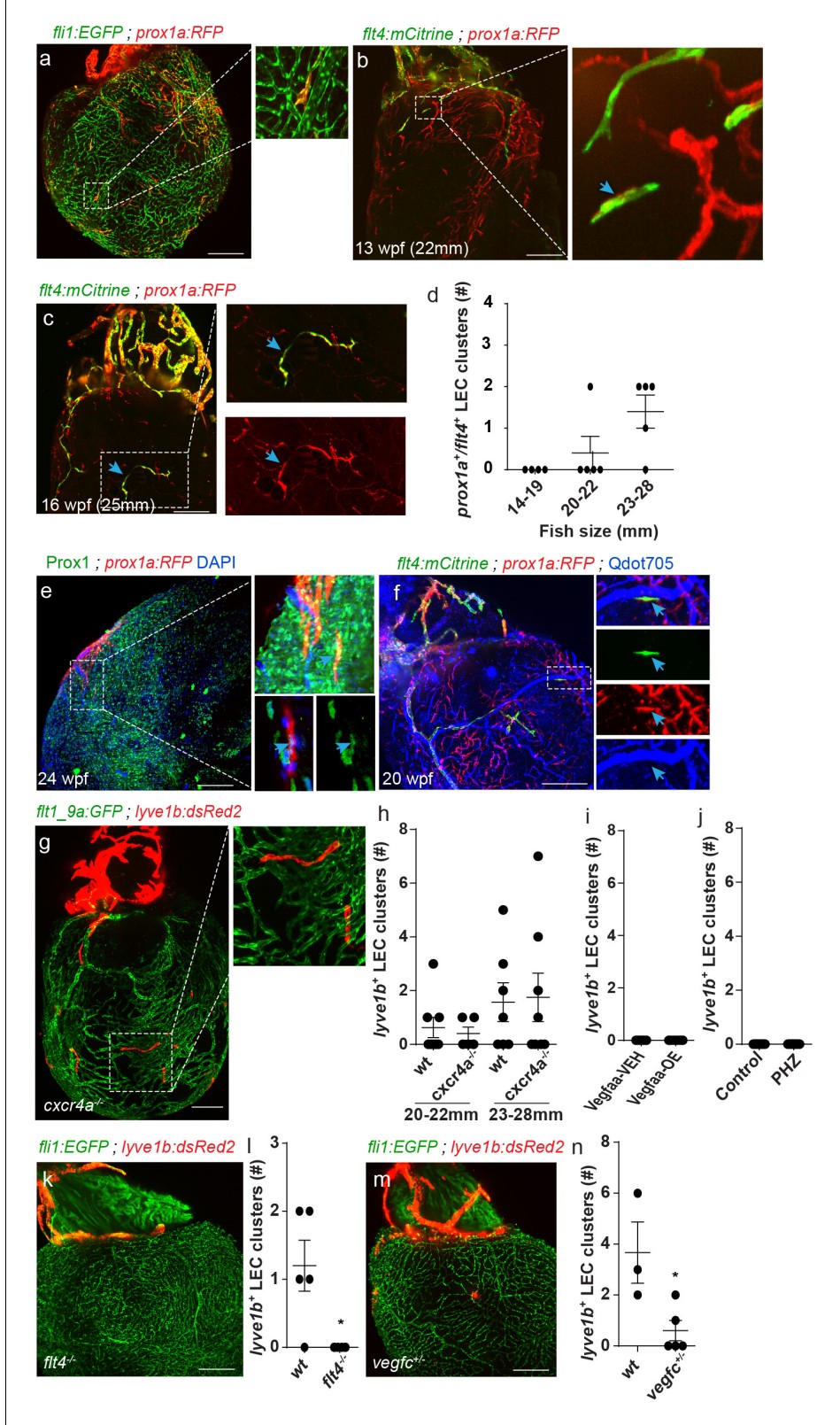

**Figure 4.** A novel population of isolated LECs is detected in the adult zebrafish heart. Insets are magnification of dashed boxes. (a) *Tg(fli1:EGFP);Tg(prox1a:KalTA4-UAS:uncTagRFP)* 16 wpf (fish size 23–28 mm) heart showing isolated lymphatic clusters (n = 8). (b,c) Double labeled *prox1a⁺;flt4⁺* isolated LECs are first detected at ~13 wpf (20–22 mm) (b, arrows) and coalesce to generate isolated capillaries by 16wpf (fish size 25–28 mm) (c, arrows). (d)

*Figure 4 continued on next page*

*Figure 4 continued*

Quantification of double-labeled *prox1a⁺;flt4⁺* isolated LECs in the ventricles of 14–28 mm fish ($n_{14-19mm}$=4, $n_{20-22mm}$=5, $n_{23-28mm}$=5). (e) *prox1a⁺* isolated LECs are also labeled by Prox1 antibody (inset, arrow). (f) 20 wpf (fish size 28 mm) double-transgenic *prox1a;flt4* hearts demonstrate that isolated LECs are not labeled following intravascular injection of Qdot705 (inset, arrow) (image in **f**) is an additional view of *Figure 2—figure supplement 1c*). (g) Isolated LEC clusters develop normally in 22 wpf (fish size 25–30 mm) *Tg(flt1_9a_cFos:GFP);Tg(lyve1b:dsRed2); cxcr4a⁻/⁻* hearts, quantified in (h) ($n_{wt20-22mm}$=8, $n_{cxcr4a^{-/-}~20-22mm}$ = 6, $n_{wt23-28mm}$=7, $n_{cxcr4a^{-/-}~23-28mm}$ = 8). (i,j) *lyve1b⁺* isolated LEC clusters are not precociously detected in Vegfaa-OE hearts (12.5 wpf, fish size 19–22 mm), in (i) ($n_{veh}$ = 7, $n_{Tam}$ = 8) or PHZ treatment (j) ($n_{control}$ = 9, $n_{PHZ}$ = 9) (k). No isolated LECs are detected in 19-23wpf (fish size 23–28 mm) *Tg(fli1:EGFP);Tg(lyve1b:dsRed2);flt4⁻/⁻* hearts, quantified in (l) ($n_{wt}$ = 5, $n_{flt4^{-/-}}$=6). (m) Significantly reduced numbers of isolated LECs are detected in *Tg(fli1:EGFP);Tg(lyve1b:dsRed2);vegfc⁺/⁻* animals at 26 wpf (fish size 25–28 mm), quantified in (n) ($n_{wt}$ = 3, $n_{vegfc^{+/-}}$=5). Scale bars are 200 μm. Error bars, mean ± s.e.m. Anterior view in a-c, e,f,k,m. Posterior view in g.

The online version of this article includes the following video and source data for figure 4:

**Source data 1.** Quantification of isolated LECs development in wt, VegfAa-OE, *flt4⁻/⁻*, *vegfc⁺/⁻*, *cxcr4a⁻/⁻* and PHZ-treated hearts.

**Figure 4—video 1.** Individual Z-stacks through heart shown in *Figure 4b* confirm there are no connections of isolated lymphatic cluster with the main ventricular lymphatic vasculature.

https://elifesciences.org/articles/44153#fig4video1

**Figure 4—video 2.** Individual Z-stacks through heart shown in *Figure 4g* confirm there are no connections of isolated lymphatic cluster with the main ventricular lymphatic vasculature.

https://elifesciences.org/articles/44153#fig4video2

**Figure 4—video 3.** Individual Z-stacks through heart of a wt sibling of *Figure 4g* confirm there are no connections of isolated lymphatic cluster with the main ventricular lymphatic vasculature.

https://elifesciences.org/articles/44153#fig4video3

---

resembles lymph-vasculogenesis (*Semo et al., 2016*). While we could not determine the exact source of these isolated LECs, we could confirm the lack of connections to ventricular lymphatics (*Figure 1f,h*; *Figure 4c,f*) and/or to the arterial network (*Figure 4f*, inset) suggesting that they may have a separate origin.

Unlike ventricular lymphatics, the isolated LEC clusters appeared normal in *cxcr4a⁻/⁻* hearts (*Figure 4g,h*, *Figure 3—figure supplement 1e*, blue arrows, *Figure 4—videos 2* and *3*). Moreover, while Vegfaa-OE induced precocious sprouting of ventricular lymphatics (*Figure 3q,r,t*) at ~12.5, when they were still absent from control hearts (*Figure 3q,t*), no effects were detected in the numbers and/or distribution of the isolated LEC following Vegfaa-OE (*Figure 4i*) or PHZ treatment (*Figure 4j*). Thus, coronaries appear to be dispensable for the formation of this lymphatic subpopulation. On the other hand, isolated LECs were completely absent from *flt4⁻/⁻* hearts (*Figure 4k,l*) and markedly reduced in *vegfc⁺/⁻* hearts (*Figure 4m,n*). Taken together, these results suggest that distinct lymphatic populations respond differently to the same molecular cues, as is the case of Cxcl12/Cxcr4 signaling which prevented ventricular lymphatic growth but did not impair the appearance of the isolated LEC clusters. In contrast, Vegfc/Flt4 signaling was absolutely required for formation of both populations, indicating its crucial role as key regulator of lymphatic development.

In order to investigate whether similar isolated LECs are also present in mammals, we examined mouse hearts stained with PROX1 and VE-Cadherin antibodies at different embryonic stages (*Figure 5a–f*). At embryonic day E13.5, coronary vessels were already present in the dorsal side of the heart (*Figure 5a*, red), but no PROX1⁺ LECs were detected. CMs were the only PROX1⁺ cells detected over the ventricle at this stage (*Figure 5a*, green). One day later at E14.5, lymphatic sprouts emanating from the sinus venosus region were clearly visualized on the dorsal aspect of the heart, consistent with previous reports (*Klotz et al., 2015*) (*Figure 5b*, white arrows). In addition to these lymphatics with clear vessel morphology, we also detected isolated PROX1⁺ LECs throughout the surface of the ventricle (*Figure 5b*, inset, blue arrows), similar to the isolated LECs found in the zebrafish heart (*Figure 4d*). Both the main lymphatic vessels and the isolated LECs also expressed VE-Cadherin. Staining with the lymphatic membrane resident receptor LYVE1 at E15.5, confirmed that these LECs clusters, positive for both PROX1 and LYVE1, were not connected to the main lymphatic vessels (*Figure 5c*, inset, arrows, *Figure 5—figure supplement 1a*). Clusters could still be found at E17.5 (*Figure 5e*, inset, arrows, *Figure 5—figure supplement 1b–c*, *Figure 5—video 1*),

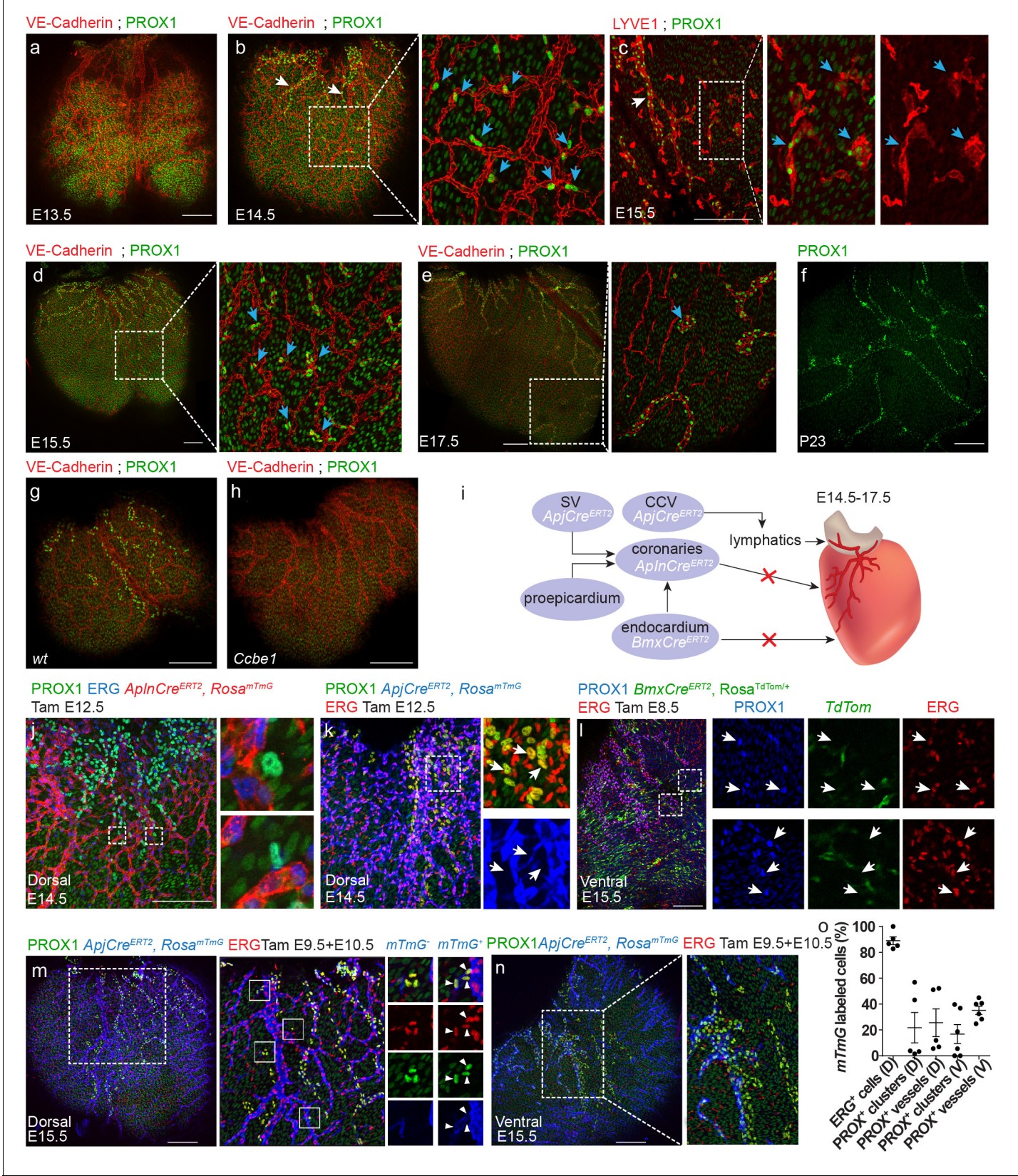

**Figure 5.** Both lymphangiogenesis and lymph-vasculogenesis contribute to cardiac lymphatic development in mammals. (a,b) Whole mount confocal images of mouse hearts immunostained for VE-Cadherin and PROX1. (a) At E13.5 coronary (red) but not lymphatic vessels (green) are present on the
*Figure 5 continued on next page*

*Figure 5 continued*

ventricle surface (n = 6) (Low levels of PROX1 are detected in cardiomyocytes). (**b**) In addition to regular lymphatic vessels (white arrows), isolated PROX1$^+$ LECs are observed in close proximity to the coronaries at E14.5 (inset, blue arrows) (n = 6). (**c**) Whole mount confocal images of E15.5 mouse hearts immunostained for LYVE1 and PROX1. Isolated PROX1$^+$/LYVE1$^+$ LECs (blue arrows) are not connected to the main lymphatic sprouts (white arrow). (**d,e**) Whole mount confocal images of mouse hearts immunostained for VE-Cadherin and PROX1. Isolated LECs gradually expand to form multicellular lumenized structures (inset, arrows) (n = 4). (**f**) PROX1$^+$ isolated LECs are no longer detected at P23 (n = 4). (**g,h**) Whole mount confocal images of E15.5 mouse hearts immunostained for LYVE1 and PROX1. PROX1$^+$ LECs are absent from *Ccbe1* mutant hearts (**h**) (n$_{wt}$ = 3, n$_{Ccbe1}$ = 3). (**i**) Lineage-tracing strategies for identification of cardiac LEC origins. (**j**) Dorsal view of *AplnCre$^{ERT2}$,Rosa$^{mTmG}$* heart from embryo dosed with tamoxifen at E12.5 and analyzed at E14.5, showing no *AplnCre$^{ERT2}$*;PROX1$^+$ LECs. Cre recombination is labeled in red, ECs in blue (Erg) and lymphatics in green (PROX1). Insets are magnification of dashed boxes. (n = 7). (**k**) Dorsal view of *ApjCre$^{ERT2}$,Rosa$^{mTmG}$* heart, showing that PROX1$^+$ LECs are not labeled by *ApjCre$^{ERT2}$* in embryos dosed with tamoxifen at E12.5 and analyzed at E14.5. Cre recombination is marked in blue, ECs in red (ERG) and lymphatics in green (PROX1). Insets are magnification of dashed boxes (n = 4). (**l**) Ventral view of heart from *BmxCre$^{ERT2}$, Rosa$^{TdTom}$* embryos dosed with tamoxifen at E8.5 and analyzed at E15.5, showing that PROX1$^+$ LECs are not labeled by *BmxCre$^{ER}$*. Cre recombination is marked in green, ECs in red (ERG) and lymphatics in blue (PROX1). Inset is magnification of dashed boxes. (n = 5). Dorsal (**m**) and ventral (**n**) views of *ApjCre$^{ERT2}$,Rosa$^{mTmG}$* embryos dosed with tamoxifen at E9.5 and 10.5 and analyzed at E15.5, showing PROX1$^+$ isolated LECs and lymphatic vessels, labeled by *ApjCre$^{ERT2}$*. Cre recombination is marked in blue, ECs in red (ERG) and lymphatics in green (PROX1). (**o**) Quantification of *mTmG* labeling shows reduced *ApjCre$^{ERT2}$* lineage traced PROX1$^+$ cells as compared to ERG ECs (recombination efficiency). D;Dorsal, V;Ventral (n$_D$ = 5, n$_V$ = 6) Insets are magnification of dashed boxes. (n$_{Dorsal}$ = 6, n$_{Ventral}$ = 5). Scale bars are 200 μm.

The online version of this article includes the following video, source data, and figure supplement(s) for figure 5:

**Source data 1.** Quantification of *ApjCre$^{ERT2}$* lineage traced PROX1$^+$ cells.
**Figure supplement 1.** Morphological and lineage analysis of mouse cardiac lymphatics.
**Figure 5—video 1.** Individual Z-stacks through LEC cluster shown in *Figure 5—figure supplement 1c* confirm there are no connections with underlying blood vessel endothelial cells.
https://elifesciences.org/articles/44153#fig5video1

---

but were no longer detected at day P23 (*Figure 5f*). Interestingly, similar to *flt4* mutant fish, *Ccbe1* mutant mice lacked both ventricular sprouts and isolated LECs (*Figure 5g,h*). Ccbe1 is required for lymphangiogenesis in mice (*Bos et al., 2011*) and fish (*Hogan et al., 2009*), acting as a regulator of VEGF-C processing (*Jha et al., 2017*). Thus, the requirement for VEGF-C/Vegfr3 signaling in both cardiac lymphatic populations is evolutionarily conserved.

We then asked what is the origin of these isolated LECs. The majority of mouse cardiac lymphatics (78%) has been shown to originate from TIE2$^+$ ECs, which emerge from the common cardinal vein (CCV) and migrate toward the heart (*Klotz et al., 2015*). Additionally, a contribution from the hemogenic endothelium has also been proposed, albeit the exact identity of these cells has remained controversial (*Klotz et al., 2015*; *Ulvmar and Mäkinen, 2016*). Individual vessels contained cells from both origins, making cardiac lymphatics a mosaic of different sources. In order to identify the cellular origins of the new population of isolated LECs, we used different lineage-tracing strategies (*Figure 5i*). Since these LEC clusters are often found in close proximity to the developing coronary vessels (*Figure 5b,d*), a feature that was also evident in the zebrafish heart (*Figure 4g*), we decided to investigate whether they originate from the local vasculature (i.e. coronary vessels). To this end, we used *AplnCre$^{ERT2}$* mice, which specifically labels sprouting ECs, but not endocardial or lymphatic cells (*Liu et al., 2015*). *AplnCre$^{ERT2}$* mice were crossed to the Cre-dependent fluorescent reporter ROSA26$^{mTmG/+}$ (*Muzumdar et al., 2007*) and Cre-mediated recombination was induced by Tamoxifen administration at day E12.5, after the coronary vasculature has formed, but prior to the appearance of lymphatics over the ventricle. Staining with PROX1 and the EC-specific ERG antibodies at E14.5 rendered no PROX1$^+$ cells that were labeled also by the *mTmG* reporter (*Figure 5j*). This result was also obtained with another Cre that labeled coronary vessels when induced at later stages, *ApjCre$^{ERT2}$* dosed at E12.5 (*Figure 5k*, inset), further confirming that LECs do not originate from the local coronary vasculature, as previously reported (*Angeli and Harvey, 2015*). Yet, we cannot exclude the possibility that ECs acquire a lymphatic fate before exiting the blood vasculature, and as a consequence, were not labeled by *AplnCre$^{ERT2}$* or *ApjCre$^{ERT2}$*.

An additional potential source is the endocardial/endothelial cells that line the lumen of the heart, which also give rise to part of the coronary blood vasculature in mouse (*Wu et al., 2012*; *Red-Horse et al., 2010*). We have recently observed that that the *BmxCre$^{ERT2}$* line is highly specific for the endocardium in the heart, when induced with Tamoxifen prior to coronary vessel development (data not shown). *BmxCre$^{ERT2}$* was originally reported as inducing recombination in embryonic

arteries specifically, but it is also highly expressed in the endocardium and almost completely excluded from the sinus venosus (*Ehling et al., 2013*). *BmxCre^ERT2* mice were crossed to ROSA26^Td-Tom/+ and Cre-mediated recombination was induced at E8.5. Hearts were harvested at day E15.5 and stained with antibodies against PROX1 and ERG. Despite almost complete endocardial recombination (data not shown), no PROX1^+;*TdTom*^+ cells were detected on the surface of the ventricle (*Figure 5l*, insets), suggesting no significant contribution of the endocardium or embryonic arteries to cardiac lymphatics. Finally, we turned back to *APJCre^ERT2* mice (*Chen et al., 2014b*), but this time induced labeling at E9.5 and E10.5 to specifically mark the major embryonic veins (SV and CCV) (*Chen et al., 2014b*) (*Figure 5—figure supplement 1d,e*). Hearts were harvested at day E15.5 and stained as described above. This manipulation resulted in lineage labeling of a portion of the lymphatic vessels and isolated clusters on both the dorsal and ventral aspects of the heart (*Figure 5m–o*), suggesting that these LECs arose from an early venous source.

Strikingly, we noted that lineage labeling was lower than expected in *APJCre^ERT2* mice dosed at E9.5/10.5 if most vessels were to derive from a venous source. We have previously shown that multiple doses of Tamoxifen result in near complete labeling of the sinus venosus (*Chen et al., 2014b*). We show here that dosing at E9.5/10.5 labels ~ 90% of sinus venosus-derived ECs on the dorsal side of the heart (*Figure 5m,o*) and a majority of the cardinal vein (*Figure 5—figure supplement 1e*). However, the relative incidence of PROX1^+/*mTmG*^+ LECs on the dorsal side of the heart was 25% within lymphatic vessels and 21% in the isolated clusters (*Figure 5m*). Similar labeling was detected on the ventral side (*Figure 5n*), with 35% PROX1^+/*mTmG*^+ cells within lymphatic vessels and 17% in the isolated clusters (*Figure 5o*). In addition, we detected a large population of lymphatics proximal to the OFT region that was not traced by *APJCre^ERT2* (*Figure 5—figure supplement 1f*), which was also reported to be negative for *Tie2Cre* labeling (*Klotz et al., 2015*). These data suggest that both ventricular lymphatics and isolated LECs are only partially derived from the *APJ*^+ lineage. Since the isolated LECs first appear as single cells associated with the blood vasculature, it is tempting to speculate that they may originate from cells carried by the blood circulation. Alternatively, they could arise from others, yet unknown non-endothelial sources.

Taken together, our results show that mammalian cardiac lymphatics form by both sprouting lymphangiogenesis and coalescence of isolated LECs (lymph-vasculogenesis). Using multiple lineage-tracing strategies, we show that neither the local coronary vasculature nor the endocardium, contribute to the different cardiac lymphatic subsets. Yet, it is important to keep in mind that despite the high recombination efficiency of the *BmxCre^ERT2*, we cannot rule out the existence of additional progenitor populations nested within the endocardium that are not labeled by this tracer.

## Differential response of cardiac lymphatics to injury

Given the importance of lymphatic vessels for immune surveillance and fluid homeostasis, it seems reasonable to hypothesize that they play similar roles during organ regeneration. Previous studies have highlighted a clear association between increased lymphatic vessel density following MI and improved cardiac function (*Klotz et al., 2015*; *Henri et al., 2016*; *Ishikawa et al., 2007*). In addition, enhancing the lymphangiogenic response after MI has been shown to augment immune cell trafficking (*Vieira et al., 2018*). Nevertheless, the cellular origins and mechanisms of response of different lymphatic subtypes to cardiac injury have not been addressed. Unlike mammals, zebrafish have a remarkable ability to regenerate their heart, making it an ideal model to study the origins and functions of the lymphatic system during organ regeneration. We employed the well-established model of cardiac cryoinjury, which closely models MI and allows visualization of the vascularization process (*González-Rosa et al., 2011*; *Marín-Juez et al., 2016*) (*Figure 6a*), to investigate the response of different cardiac lymphatic subsets to injury. Injuries were performed on 6–18 mpf (fish size 25–32 mm) transgenic zebrafish, hearts were harvested at 40 hr post cryoinjury (hpci) or at 7, 14, 21, and 73 days post cryoinjury (dpci). Notably, we found that *prox1a*+ sprouts (*Figure 6b,c*, white arrows), as well as new *prox1a*+ isolated LECs (*Figure 6b,c*, yellow arrows) were the first to be detected in the injured area, as early as at ~40 hpci. While part of these sprouts were also labeled by the blood EC marker *flt1_9a* (*Figure 6c* insets, white arrows) and may represent blood arterioles (*Marín-Juez et al., 2016*), the isolated LECs in the injured area were not marked by the arterial-specific transgene (*Figure 6c*, insets, yellow arrows), supporting their lymphatic identity. As a whole, the large majority of lymphatics in the injured area between 40 hpci-14 dpci, were not connected to large collecting lymphatics of the OFT, or to ventricular lymphatics (*Figure 6d–i*, *Figure 6—figure*

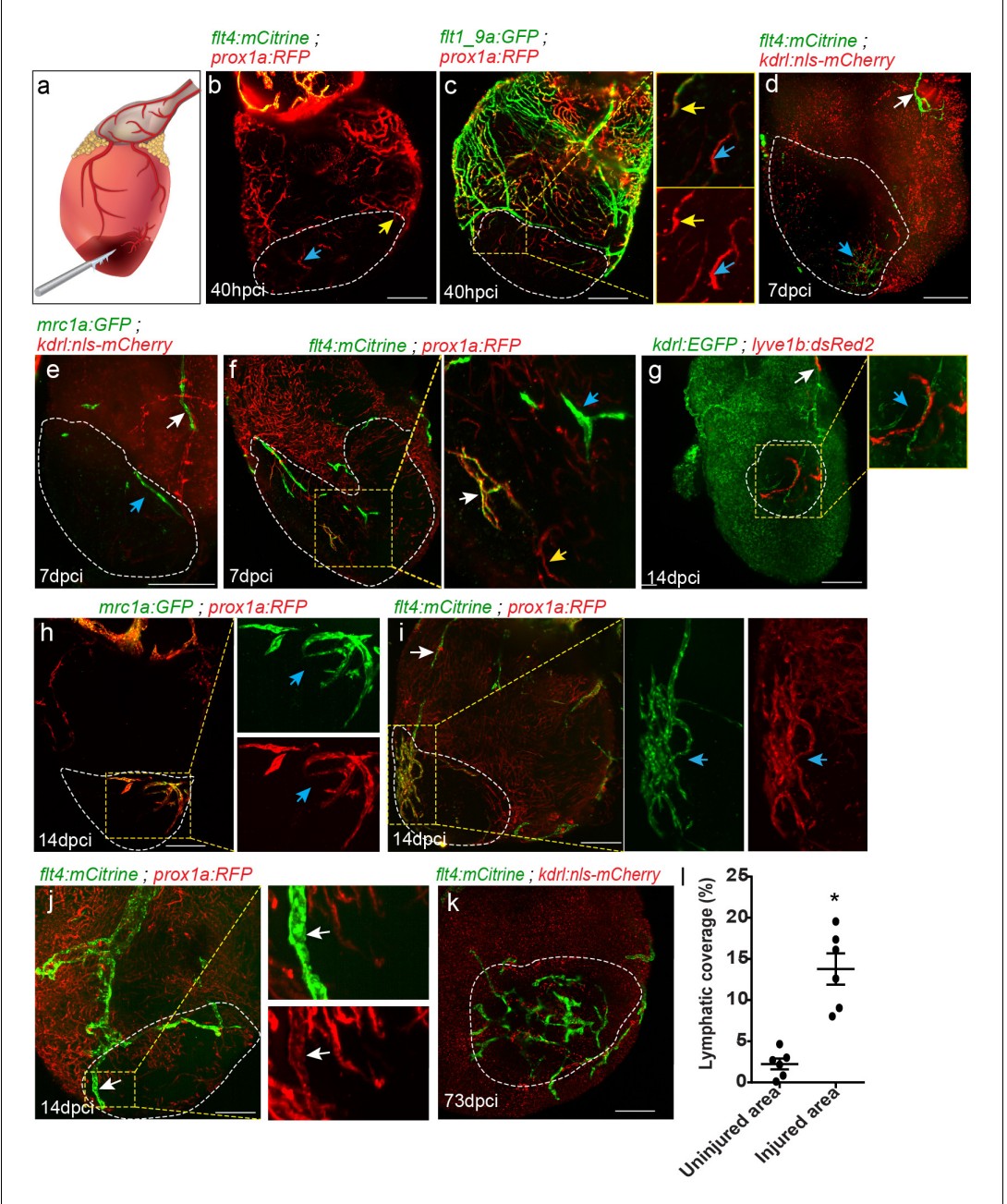

**Figure 6.** Differential response of cardiac lymphatics to injury. (a) Diagram depicting the cryoinjury procedure. Injured area is outlined in all images, insets show high-magnification of dashed boxes. (b) *flt4;prox1a* transgenic hearts at 40 hpci showing *prox1a+* sprouts (yellow arrow) and isolated LECs (blue arrow) in the injured area (n = 5). (c) *Tg(flt1_9a_cFos:GFP);Tg(prox1a:KalTA4-UAS:uncTagRFP)* double labeled coronary sprouts (inset, yellow arrow), as well as *prox1a+* isolated LECs (inset, blue arrow) are detected in the injured area at 40 hpci. (d–f) seven dpci injured hearts of (d) *Tg(kdrl:nls-mCherry);Tg(flt4BAC:mCitrine)* (n = 5), (e) *Tg(kdrl:nls-mCherry);Tg(mrc1a:EGFP)* (n = 5), and (f) *Tg(prox1a:KalTA4-UAS:uncTagRFP);Tg(flt4BAC:mCitrine)* (n = 5) fish, with white arrows pointing to OFT-connected ventricular lymphatics, and blue arrows pointing to isolated LECs in the injured area. (g–i) 14 dpci ventricles of (g) *Tg(flt1_9a_cFos:GFP;Tg(lyve1b:dsRed2)* (n = 8), (h) *Tg(prox1a:KalTA4-UAS:uncTagRFP)* (n = 5) and (i) *Tg(prox1a:KalTA4-UAS:uncTagRFP);Tg(flt4BAC:mCitrine)* (n = 5) showing isolated lymphatic sprouts in the injured area (blue arrows), which are not connected to ventricular lymphatics (white arrow) (j) Double labeled *prox1a;flt4* ventricular lymphatic sprouts invade the injured area at 14 dpci (inset, white arrows) (n = 3). (k) 73 dpci *Tg(kdrl:nls-mCherry);Tg(flt4BAC:mCitrine)* heart showing increased lymphatic coverage in the injured vs. uninjured areas of the ventricle, quantified in (l). Error bars, mean ± s.e.m. *p<0.001. Scale bars are 200 μm. Fish size 25–30 mm.

The online version of this article includes the following video, source data, and figure supplement(s) for figure 6:

**Source data 1.** Quantification of lymphatic coverage 73 dpci.

**Figure supplement 1.** Lymph-vasculogenesis represents the main mechanism of lymphatic formation after cardiac injury.

*Figure 6 continued on next page*

Figure 6 continued

**Figure 6—video 1.** Individual z-stacks through heart shown in *Figure 6f* confirm there are no connections of isolated lymphatic cluster with the main ventricular lymphatic vasculature.

https://elifesciences.org/articles/44153#fig6video1

**Figure 6—video 2.** Individual z-stacks through heart shown in *Figure 6g* confirm there are no connections of isolated lymphatic cluster with the main ventricular lymphatic vasculature.

https://elifesciences.org/articles/44153#fig6video2

**Figure 6—video 3.** Individual z-stacks through heart shown in *Figure 6h* confirm there are no connections of isolated lymphatic cluster with the main ventricular lymphatic vasculature.

https://elifesciences.org/articles/44153#fig6video3

**Figure 6—video 4.** Individual Z-stacks through heart shown in *Figure 6—figure supplement 1h* confirm there are no connections of isolated lymphatic cluster with the main ventricular lymphatic vasculature.

https://elifesciences.org/articles/44153#fig6video4

supplement 1a,b,d,f). While we could not determine the origins of the injury-specific LEC clusters, which did not appear in sham operated hearts (*Figure 6—figure supplement 1c,e,g*), they were highly heterogeneous both in morphology and gene expression. We could detect cells expressing combinations of LEC markers, such as *prox1a* and *flt4* (*Figure 6f*, inset, white arrow, *Figure 6—video 1*), as well as cells expressing *prox1a* (*Figure 6f*) or *flt4* only (*Figure 6f*, inset, blue arrow). Interestingly, isolated LECs in the injured area developed into long sprouts (*Figure 6h–j*, inset, *Figure 6—video 3*), which continued growing throughout the regenerative phase and could be detected even after 2 months pci (*Figure 6l,m*, *Figure 6—figure supplement 1g,h*, *Figure 6—video 2*).

Contrary to the rapid lymph-vasculogenic response, OFT-derived ventricular lymphatics reached the regenerating area only at ~1–3 weeks pci (*Figure 6j*, arrows), mostly following their own-damage, as lymphatic vessels that were remote from the injury site, did not respond at all (data not shown).

We then asked whether the same cues controlling the development of cardiac lymphatics are reactivated during injury. In line with the reduction in ventricular lymphatics (*Figure 3m–p*) and isolated lymphatic clusters (*Figure 4j–m*) observed in $flt4^{-/-}$ and $vegfc^{+/-}$ hearts, these were nearly absent in the vicinity, as well as within the injured area of ~1 month post-cryoinjury (mpci) mutant hearts (*Figure 7a–f*, arrows). Even at ~2 mpci, when an extensive lymphatic network covered the injury in wt hearts, $vegfc^{+/-}$ hearts remained largely devoid of lymphatic vessels (*Figure 7g–i*, arrows). In line with the impaired post-injury neo-lymphangiogenesis (*Figure 7a–i*), the scar area was increased in injured $flt4^{-/-}$ and $vegfc^{+/-}$ hearts (*Figure 7j–o*, arrow). Acid Fuchsin Orange-G (AFOG) staining of heart sections showed that while in control hearts the fibrotic scar was mostly resolved by ~1 mpci and replaced by new muscle tissue, $flt4^{-/-}$ hearts displayed a prominent scar composed of a collagenous core and a thin layer of fibrin at the border of the wound (*Figure 7j–l*, arrow). Similarly, the scar was not resolved in $vegfc^{+/-}$ hearts even at ~2 mpci (*Figure 7m–o*, arrow).

In contrast to these results, isolated lymphatic clusters, which develop normally in $cxcr4^{-/-}$ hearts (*Figure 4h–i*), could be readily detected following injury as well (*Figure 7p–r*), suggesting that similar programs control lymphatic growth during development and repair. Nevertheless, scar area was increased in $cxcr4^{-/-}$ mutant hearts at 77 dpci (*Figure 7s–u*, arrow), supporting previous studies suggesting a cell-autonomous requirement of Cxcl12/Cxcr4 signaling for CM and/or coronary migration (*Harrison et al., 2015*; *Itou et al., 2012*).

## Discussion

In this study, we investigated the mechanisms underlying the development of the cardiac lymphatic system in zebrafish and mouse, and analyzed the response of the lymphatic system to cardiac injury. We demonstrate that the cardiac lymphatic vasculature is composed of distinct subsets, which respond differently to lymphangiogenic cues and display unique behaviors during cardiac injury and regeneration. Our results suggest that cardiac injury induces the de novo formation of lymphatics, through a process reminiscent of lymph-vasculogenesis rather than sprouting lymphangiogenesis. The heterogeneous molecular nature of the newly formed lymphatic sprouts, combined with the fact

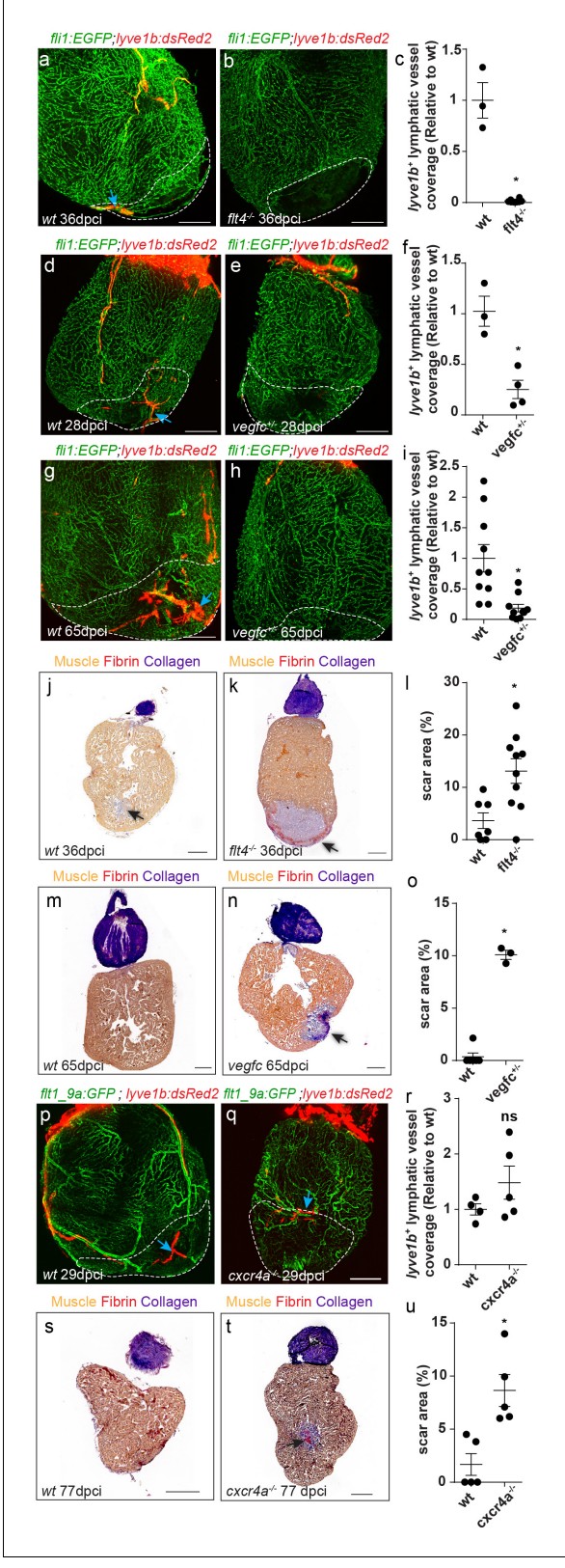

**Figure 7.** cardiac regeneration is impaired in *flt4/vegfc* mutant fish. Injured area is outlined in all images (**a,b**) *lyve1b*+ lymphatic capillaries (blue arrows) are detected in the injured area of 36 dpci *Tg(fli1:EGFP); Tg(lyve1b: dsRed2)* wt sibling (**a**) but not in *flt4*-/- hearts (**b**). (**c**) Quantification of *lyve1b*+ lymphatic vessel coverage in the injured area of 35–38 dpi *flt4*-/- hearts ($n_{wt}$ = 3, $n_{flt4}$-/-=8, *p<0.001). (**d–i**) *lyve1b*+ lymphatic capillaries (blue arrows)

*Figure 7 continued on next page*

*Figure 7 continued*

are detected in the injured area of 28 dpci (**d**) and 65 dpci (**g**) *Tg(fli1:EGFP); Tg(lyve1b:dsRed2)* wt siblings, but not in *vegfc*$^{+/-}$ heterozygous fish (**e,h**). (**f,i**) Quantification of *lyve1b*$^+$ lymphatic vessel coverage in the injured area of 21–28 dpi *vegfc*$^{+/-}$ hearts ($n_{wt}$ = 3 hearts, $n_{vegfc}$ $^{+/-}$=4, *$p<0.005$) and 65–66 dpi *vegfc* $^{+/-}$ hearts ($n_{wt}$ = 10 hearts, $n_{vegfc}$$^{+/-}$=10, *$p<0.005$). (**j,k**) AFOG-stained sections at 36 dpci showing lack of regeneration in *flt4* $^{-/-}$ hearts (**k**) as compared to wt siblings (**j**). Collagenous scar is stained in blue, fibrin in red, and cardiac muscle in orange. Black arrow points to scar. (**l**) Increased scar area (calculated as percent of ventricle) in *flt4*$^{-/-}$ hearts ($n_{wt}$ = 8 hearts, $n_{flt4}$$^{-/-}$=11, *$p<0.05$). (**m,n**) AFOG-stained sections at 65 dpci showing lack of regeneration in *vegfc*$^{+/-}$ hearts (**n**) as compared to wt siblings (**m**). Black arrow points to scar. (**o**) Increased scar area (calculated as percent of ventricle) in *vegfc*$^{+/-}$ hearts ($n_{wt}$ = 6 hearts, $n_{vegfc}$$^{+/-}$=3, *$p<0.001$). (**p,q**) *lyve1b*$^+$ isolated LEC clusters (blue arrows) are normally detected in the injured area of 29 dpci *Tg(flt1_9a_cFos:GFP);Tg(lyve1b:dsRed);cxcr4a*$^{-/-}$ (**q**) and wt sibling (**p**) hearts. (**r**) Quantification of *lyve1b*$^+$ lymphatic vessel coverage (relative to wt sibling) in 29 dpi *cxcr4a* $^{-/-}$ hearts ($n_{wt}$ = 4 hearts, $n_{cxcr4a}$$^{-/-}$=5). (**s,t**) AFOG-stained sections at 77 dpci showing impaired regeneration in *cxcr4a*$^{-/-}$ mutant (**t**) as compared to wt siblings (**s**). Black arrow points to scar. (**u**) Increased scar area (calculated as percent of ventricle) in *cxcr4a*$^{-/-}$ mutant hearts ($n_{wt}$ = 5 hearts, $n_{cxcr4a}$$^{-/-}$=5, *$p<0.005$). Scale bars are 200 µm. All fish size are 25–30 mm.

The online version of this article includes the following video and source data for figure 7:

**Source data 1.** Quantification of lymphatic coverage and scar area in cryoinjured *flt4* $^{-/-}$, *vegfc*$^{+/-}$ and *cxcr4a* $^{-/-}$ hearts.
**Figure 7—video 1.** Individual Z-stacks through heart shown in *Figure 7q* confirm there are no connections of isolated lymphatic cluster with the main ventricular lymphatic vasculature.
https://elifesciences.org/articles/44153#fig7video1

that they are initially not connected to the pre-existing cardiac lymphatic vasculature, may reflect on alternative LEC origins, specifically activated in response to injury. Finally, our results indicate that neo-lymphangiogenesis following cardiac injury is necessary for proper scar resolution and cardiac regeneration.

Recent studies have reported the presence of isolated LECs in different organs during embryogenesis. In zebrafish for instance, a novel population of perivascular LECs was found in the meninx. These cells differentiate into dispersed, non-lumenized structures that act as scavenger cells and persist throughout life (*Bower et al., 2017b*; *van Lessen et al., 2017*). In the mouse, isolated LECs have been shown to contribute to lymphatic vessels in the skin (*Martinez-Corral et al., 2015*; *Pichol-Thievend et al., 2018*), the mesentery (*Stanczuk et al., 2015*) and most recently also the intracranial and spinal meninges (*Antila et al., 2017*). We find that in the mouse heart, isolated LECs represent transient structures that are no longer detected by P23, once the lymphatic system is fully formed. In zebrafish on the other hand, the presence of isolated LECs is detected at all stages, most probably correlating with the late onset and continuous growth of cardiac lymphatics throughout adult life. Moreover, these clusters may serve as an available source of LECs to support lymphatic growth in response to cardiac injury or other tissue needs.

Our analyses of the cellular origins of cardiac lymphatics in zebrafish and mice provide novel insights into the ontogeny of this system. We find that in zebrafish, the OFT lymphatics along with their derived ventricular lymphatics, originate in the VFL (*Figure 2*), a late forming facial lymphatic vessel. Interestingly, the zebrafish facial lymphatic network itself has recently been shown to form from three progenitor populations. Two of them are of venous origins (CCV and PHS) while the third one is angioblast-derived, and contributes not only to facial lymphatics but also to the hypobranchial artery (*Eng et al., 2019*). Our results therefore, indicating that OFT lymphatics sprout from facial lymphatics and not from the trunk vasculature, rise the appealing hypothesis that certain components of the heart, such as the OFT, along with their accompanying lymphatic vasculature could share common cardiopharyngeal mesoderm (CPM) origins (reviewed in *Diogo et al., 2015*). This idea, receives further support from recent lineage-tracing experiments in mice revealing that a population of *Isl1*-expressing CPM progenitors contribute LECs to the ventral side of the ventricles and the OFT (*Maruyama et al., 2019*).

The multiple spatiotemporally controlled lineage-tracing strategies revealed that in mice, both lymphatic vessels and isolated clusters have, at least in part, a venous origin. These two LEC populations may originate from different veins (such as the cardinal vein and sinus venosus). Alternatively, they may arise from the same vein, employing different mechanisms of migration and sprouting to

reach the heart, thus ensuring proper and perhaps faster lymphatic coverage. Finally, it could also be possible that some cells detach from the parent lymphatic vessel and migrate through as an isolated cluster, as it has been shown for LECs in the lung, which bud from extra-pulmonary lymphatics and migrate as single cells or small clusters into the developing lung (*Kulkarni et al., 2011*).

Notably, the relative low fraction of LECs labeled by *ApjCre^ERT2* (20–30%), compared to the high recombination efficiency of this Cre driver (~90%), hints at putative additional sources with major contribution to both isolated and 'traditional' cardiac lymphatics. One possibility is that the isolated LEC clusters originate from non-venous migratory progenitor cells as previously shown for other organs (*Martinez-Corral et al., 2015*; *Stanczuk et al., 2015*). Most recently, yolk sac born erythro-myeloid progenitors (EMPs) were shown to contribute to blood ECs in multiple organs, including the heart (*Plein et al., 2018*), raising the possibility that they could also contribute to lymphatic ECs in the heart. Finally, whether the isolated LEC clusters in mouse and zebrafish derive from the CPM remains an open question.

While debate continues over the venous vs. non-venous origins of cardiac lymphatics: a large proportion of cardiac LECs has recently been traced by *Pax3Cre*, a well-established marker of the dermomyotome (a subset of the somitic paraxial mesoderm). In this case, PROX1-expressing LEC precursors could be traced by *Pax3Cre* while still located in the dorsolateral wall of the CV (*Engleka et al., 2005*), suggesting that even the venous-derived cardiac lymphatics could be more heterogeneous than previously appreciated. In the future, it would be interesting to ascertain whether specific molecular characteristics or functions are ascribed to LECs of distinct origins.

Our results uncover significant differences in the series of events leading to the establishment of distinct lymphatic populations within the heart, and exemplify how each lymphatic subset responds differently to the same molecular cues. We find that the upregulation of pro-lymphangiogenic cues in the OFT, highly correlates with changes in heart rate associated with larva to juvenile transition. Hence, the increased demand for fluid drainage appears to elicit a lymphangiogenic response leading to the formation of OFT lymphatics. Interestingly, while lymphatics colonize the OFT prior to the appearance of the blood vasculature, ventricular lymphatics closely follow coronary vessels in both fish and mice, and fail to sprout in zebrafish *cxcr4a* mutants, which bear an immature coronary plexus. These results, along with the correlated increase in blood and lymphatic vessel growth induced by PHZ and Vegfaa-OE, strongly support a role for the coronary vasculature in ventricular lymphatic growth. We further show that lymphatics fail to form in zebrafish *vegfc* mutants, even when hearts are hyper-vascularized as a result of Vegfaa-OE, suggesting that a mature coronary plexus may act as a scaffold for ventricular lymphatic development, in a Vegfc-dependent manner. In contrast to ventricular lymphatics, the isolated LEC clusters are spread throughout the ventricle, and do not follow a specific path. Therefore, their formation was not affected by changes in the coronary vasculature, that is they were normally detected in *cxcr4a* mutant hearts and were not precociously induced by Vegfaa-OE or PHZ treatment (*Figure 4*). Yet, this population was fully absent from *flt4^-/-* and *vegfc^+/-* hearts, pointing to an additional source, other than coronary ECs, for Vegfc. While Vegfc has been previously shown to be expressed in maturing coronary artery ECs (*Su et al., 2018*), supporting a role of blood vessels in guiding lymphatic growth, *Vegfc* and *Ccbe1* (*Bonet et al., 2018*) expression has also been detected in the epicardium in mice (*Chen et al., 2014b*), as well as in fish CMs under certain conditions (*Marín-Juez et al., 2016*). As a whole, the formation of the different lymphatic subsets in zebrafish, clearly follows the tissue requirements. OFT lymphatics develop during larva to juvenile transition in response to hemodynamic changes eliciting the demand for fluid drainage. Later on, ventricular lymphatics sprouts in response to cardiac muscle expansion and coronary formation. Finally, the LEC clusters emerge and incorporate into a growing capillary network, thereby increasing myocardial lymphatic coverage.

While previous reports have described an increase in lymphatic vessel density following MI (*Klotz et al., 2015*; *Henri et al., 2016*), little is known about the cellular and molecular mechanisms by which cardiac lymphatics form in response to injury. Interestingly, we find that the majority of lymphatics in the zebrafish regenerating area form de novo, whereas only a few arise from pre-existing ventricular lymphatics. In similar fashion, recruitment of isolated LECs was shown to take place during wound healing in the adult mouse skin, which later interconnect through a process reminiscent of lymph-vasculogenesis (*Boardman and Swartz, 2003*; *Rutkowski et al., 2006*). In contrast, at the periphery of the wound, lymphangiogenesis occurs by sprouting of pre-existing lymphatic vessels (*Paavonen et al., 2000*). As wound healing and regeneration share common features, including an

immediate inflammatory response, revascularization, innervation and formation of a fibrotic scar (*Richardson, 2018*) it seems reasonable to speculate that similar mechanisms may control the formation of lymphatics in both of these processes. Nonetheless, our results indicate that all lymphatics in the injured area derive from intra-cardiac sources. These could include the isolated LECs, or LECs detaching from ventricular lymphatics, as recently shown for collateral artery assembly following MI in mice (*Das et al., 2019*). Another potential source is the endocardium. A subset of hemogenic angioblasts, expressing cardiac markers has been previously shown to contribute to the endocardium and to serve as a source for transient definitive hematopoietic progenitors (*Nakano et al., 2013*; *Zamir et al., 2017*). Moreover, the endocardium itself was found to possess angiogenic capabilities (*Wu et al., 2012*; *Chen et al., 2014b*). While we could not detect definitive endocardial contribution to developing cardiac lymphatics (*Figure 5*), the endocardium may hold the ability to give rise to LECs under pathological conditions.

Regardless of their origin, studies emerging in recent years have ascribed a beneficial role for lymphangiogenesis promoting therapies in different aspects of MI pathology, including cardiac edema, inflammation and scarring (*Klotz et al., 2015*; *Henri et al., 2016*; *Vieira et al., 2018*). Moreover, exogenous supply of VEGF-C has been shown to improve cardiac function after MI in mice (*Klotz et al., 2015*). Here, we show that an intact lymphatic system is instrumental for cardiac regeneration in zebrafish, as *flt4* and *vegfc$^{+/-}$* animals fail to regenerate their hearts, despite the presence of a normal coronary vasculature (*Figure 7*). While we cannot exclude the possibility that Vegfc/Flt4 signaling is autonomously required in other cell populations (e.g. blood ECs and/or macrophages), our data point to the absence of lymphatic vessels as the main reason precluding heart regeneration.

Most recently, an additional study has also examined cardiac lymphatics in zebrafish carrying a hypomorphic allele of *vegfc* (*Le Guen et al., 2014*) and a null mutation in *vegfd* (*Bower et al., 2017a*). While double heterozygote animals showed a marked reduction in ventricular lymphatics similar to the *vegfc$^{+/-}$* mutants used in this study, double mutants were almost completely devoid of cardiac lymphatics, as were the *flt4$^{-/-}$* fish described herein. Strikingly, cardiac regeneration was normal in most of the *vegfc$^{+/-}$;vegfd$^{-/-}$* double mutant hearts, with only some displaying impaired regeneration capacities (*Vivien et al., 2019*). While the penetrance of the phenotypes described by Vivien et al. was lower than the observed in this study, perhaps due to the use of different mutant alleles, both studies describe a robust lymphangiogenic response to cardiac cryoinjury and point to a role for cardiac lymphatics in cardiac regeneration. In the future, further analyses will be required in order to ascertain what is the precise function that the different lymphatic subsets play in cardiac regeneration.

# Materials and methods

## Key resources table

| Reagent type (species) or resource | Designation | Source or reference | Identifiers | Additional information |
|---|---|---|---|---|
| Strain (*Danio rerio*) | Tg(fli1:EGFP)$^{yl}$ | (*Nicenboim et al., 2015*) | ZDB-ALT-011017–8 | |
| Strain (*Danio rerio*) | Tg(lyve1b:dsRed2)$^{nz101}$ | (*Nicenboim et al., 2015*) | ZDB-ALT-120723–3 | |
| Strain (*Danio rerio*) | Tg(gata1a:dsRed)$^{sd2}$ | (*Nicenboim et al., 2015*) | ZDB-ALT-051223–6 | |
| Strain (*Danio rerio*) | TgBAC(prox1a: KalTA4-4xUAS-E1b: uncTagRFP)$^{nim5}$ | (*Nicenboim et al., 2015*) | ZDB-ALT-140521–3 | |
| Strain (*Danio rerio*) | Tg(flt1_9a_cFos:GFP)$^{wz2}$ | (*Nicenboim et al., 2015*) | ZDB-ALT-150723–14 | |
| Strain (*Danio rerio*) | Tg(flt4BAC:mCitrine)$^{hu7135}$ | (*van Impel et al., 2014*) | ZDB-ALT-140521–1 | |

*Continued on next page*

*Continued*

| Reagent type (species) or resource | Designation | Source or reference | Identifiers | Additional information |
|---|---|---|---|---|
| Strain (*Danio rerio*) | Tg(mrc1a:EGFP)$^{y251}$ | (*Jung et al., 2017*) | ZDB-ALT-170717–2 | |
| Strain (*Danio rerio*) | Tg(kdrl:nls-mCherry)$^{y173}$ | (*Fujita et al., 2011*) | ZDB-ALT-110429–4 | |
| Strain (*Danio rerio*) | Tg(kdrl:EGFP)$^{s843}$ | (*Jin et al., 2005*) | ZDB-ALT-050916–14 | |
| Strain (*Danio rerio*) | cxcr4a$^{um20}$ | (*Siekmann et al., 2009*) | ZDB-ALT-091124–1 | |
| Strain (*Danio rerio*) | cxcl12b$^{mu100}$ | (*Bussmann et al., 2011*) | ZDB-ALT-110513–2 | |
| Strain (*Danio rerio*) | vegfc$^{um18}$ | (*Villefranc et al., 2013*) | ZDB-ALT-130718–3 | |
| Strain (*Danio rerio*) | Tg(βactin2:loxP-mTagBFP-STOP-loxP-vegfaa)$^{pd262}$; Tg(cmlc2:CreER) | (*Karra et al., 2018*) | ZDB-ALT-181129–18 | |
| Strain (*Danio rerio*) | flt4$^{um203}$ | (*Kok et al., 2015*) | ZDB-ALT-160721–30 | |
| Strain (*Danio rerio*) | vegfd$^{bns257}$ | This paper | N/A | CRISPR/Cas9 generated. Prof. Didier YR Stainier (Max Planck Institute for Heart and Lung Research, Germany) |
| Strain (*Mus musculus*) | CD1 (wild type) | Charles River Laboratories | Strain# 022 | |
| Strain (*Mus musculus*) | FVB (wild type) | Charles River Laboratories | Strain# 207 | |
| Strain (*Mus musculus*) | ApjCre$^{ERT2}$ | (*Chen et al., 2014b*) | MGI:5689869 | |
| Strain (*Mus musculus*) | BmxCre$^{ERT2}$ | (*Ehling et al., 2013*) | MGI:5513853 | |
| Strain (*Mus musculus*) | AplnCre$^{ERT2}$ | (*Liu et al., 2015*) | MGI:5637737 | |
| Strain (*Mus musculus*) | Ccbe1 | (*Bos et al., 2011*) | N/A | |
| Strain (*Mus musculus*) | Rosa$^{mTmG}$ | (*Muzumdar et al., 2007*) | Stock# 007676 | |
| Strain (*Mus musculus*) | Rosa$^{TdTomato}$ | (*Muzumdar et al., 2007*) | Stock #007909 | |
| Sequence-based reagent | cxcl12a_F | This paper | PCR primers | CGTAGTAGTCGCT CTGATGG |
| Sequence-based reagent | cxcl12a_R | This paper | PCR primers | TGGGACTGTGTTG ACTGTGGAA |
| Sequence-based reagent | cxcl12b_F | This paper | PCR primers | GGAGCATCCGAGA GATCAAG |
| Sequence-based reagent | cxcl12b_R | This paper | PCR primers | TGTTCTTCAGCTT GGCAATG |
| Sequence-based reagent | Vegfc_F | (*Astin et al., 2014*) | PCR primers | AAGGGCCCTAACA GAATGTC |
| Sequence-based reagent | Vegfc_R | (*Astin et al., 2014*) | PCR primers | TTTGAATGAAGGG TGTCAGG |
| Antibody | anti-PROX1 (Rabbit polyclonal) | Abcam | Cat# 11941 | IF(1:700) |

*Continued on next page*

*Continued*

| Reagent type (species) or resource | Designation | Source or reference | Identifiers | Additional information |
|---|---|---|---|---|
| Antibody | anti-VE-Cadherin (Rat polyclonal) | BD Pharmingen | Cat# 550548 | IF(1:100) |
| Antibody | anti-PROX1 (Goat polyclonal) | R and D Systems | Cat#: AF2727 | IF(1:300) |
| Antibody | anti- ERG (Rabbit monoclonal) | Abcam | Cat#: ab92513 | IF(1:1000) |
| Antibody | anti- LYVE-1 (Rat monoclonal) | eBiosciences | Cat#: 14-0443-80 | IF(1:100) |
| Antibody | Alexa Fluor Conjugated Secondary Antibodies (488,594,633,635,647) | Life Technologies | N/A | IF(1:250) |
| Chemical compound, drug | Atenolol | Sigma Aldrich | A7655 | |
| Chemical compound, drug | Phenylhydrazine hydrochloride (PHZ) | Sigma Aldrich | 78690 | |
| Chemical compound, drug | 4-hydroxytamoxifen | Sigma Aldrich | H7904 | |
| Commercial assay or kit | Qtracker705 | Invitrogen | Q21061MP | |
| Commercial assay or kit | Acid Fuchsin Orange-G (AFOG) | DIAPATH | 010307 | |
| Software, algorithm | Angiotool | (*Zudaire et al., 2011*) | N/A | |
| Software, algorithm | Image J | NIH (https://www.nih.gov/ij/) | N/A | |

## Zebrafish husbandry and transgenic lines

Zebrafish were raised by standard methods (*Gibbs-Bar et al., 2016*) and handled according to the guidelines of the Weizmann Institute Animal Care and Use Committee. Zebrafish lines used in this study were: $Tg(fli1:EGFP)^{yl}$, $Tg(lyve1b:dsRed2)^{nz101}$, $Tg(gata1a:dsRed)^{sd2}$, $TgBAC(prox1a:KalTA4-4xUAS-E1b:uncTagRFP)^{nim5}$ and $Tg(flt1\_9a\_cFos:GFP)^{wz2}$ (*Nicenboim et al., 2015*); $Tg(flt4^{BAC}:mCitrine)^{hu7135}$ (*van Impel et al., 2014*), $Tg(mrc1a:EGFP)^{y251}$ (*Jung et al., 2017*), $Tg(kdrl:nls-mCherry)^{y173}$ (*Fujita et al., 2011*), $Tg(kdrl:EGFP)^{s843}$ (*Jin et al., 2005*), $Tg(myl7:GFP)$ (*González-Rosa et al., 2011*), $cxcr4a^{um20}$ (*Siekmann et al., 2009*), $cxcl12b^{mu100}$ (*Bussmann et al., 2011*), $vegfc^{um18}$ (*Villefranc et al., 2013*) $flt4^{um203}$ (*Kok et al., 2015*), $Tg(\beta actin2:loxP-mTagBFP-STOP-loxP-vegfaa)^{pd262}$; $Tg(cmlc2:CreER)$ (*Karra et al., 2018*). $vegfd^{bns257}$ mutants were generated by targeted genome editing using the CRISPR/Cas9 system as previously described (*Matsuoka et al., 2016*). The second exon of *vegfd*, which encodes part of the Vegfd protein coding sequences prior to the Vegf homology domain was targeted. The $vegfd^{bns257}$ mutant allele harbors a 59 base-pair insertion in the exon 2 of *vegfd* predicted to lead to a premature stop codon at tyrosine residue 56 thus yielding a truncated polypeptide containing a stretch of 55 amino acids of Vegfd. The following guide RNA (gRNA) sequence was used to target the exon 2 of *vegfd*: 5'- GATGTTGACCGAATACC −3'. 1 nl of a solution containing 250 ng/µl of Cas9 mRNA and 100 ng/µl of gRNA was injected at the one-cell stage. Wt and $vegfd^{bns257}$ animals were identified by PCR using the following primers at expected PCR product sizes (127 bp for wt allele and 186 bp for $vegfd^{bns257}$ allele): $vegfd^{bns257}$ forward primer: 5'- GACACAAATCAGGAAAAGTGG −3' $vegfd^{bns257}$ reverse primer: 5'- CATCGAAG TGCTTCAGCTTG −3'

Experiments were conducted on fish from the same clutch, which were of same age (weeks post-fertilization), and size -based on standard body length (the distance from the snout to the caudal peduncle) (*Parichy et al., 2009*). Initially fish were selected based on their age (~22 dpf VFL development, ~3–8 wpf OFT lymphatics development, ~10–20 wpf OFT lymphatics in mutants, ~22–

24 wpf ventricular lymphatics). Subsequently, they were anesthetized by immersion into 0.04% tricaine and placed on a ruler for measuring standard body length. Fish above or below a certain size range, as stated in the text, were excluded from the experiment.

For imaging of up to four wpf larvae, embryos were either treated for 7 days with 0.003% N-phenylthiourea (PTU) (Sigma, St Louis, MO) to inhibit pigment formation or casper (*roy*$^{-/-}$;*nacre*$^{-/-}$) mutant fish were used (*White et al., 2008*).

## Mice

Mouse use followed Stanford IACUC guidelines. Strains used were wild type (CD1 and FVB, Charles River Laboratories), *ApjCre*$^{ERT2}$ (51), *BmxCre*$^{ERT2}$ (*Ehling et al., 2013*), *AplnCre*$^{ERT2}$ (*Liu et al., 2015*) and *Ccbe1* (*Bos et al., 2011*).

Cre lines were crossed to Rosa$^{mTmG}$ or to Rosa$^{TdTomato}$ (*Muzumdar et al., 2007*). Pregnant females were dosed intraperitoneally (4 mg of tamoxifen in corn oil at E8.5 for *BmxCre*$^{ERT2}$, E9.5+E10.5 *ApjCre*$^{ERT2}$ or E12.5 for *AplnCre*$^{ERT2}$ and *ApjCre*$^{ERT2}$). Hearts were analyzed at E13.5-P23.

## Cryoinjury

Cryoinjury was performed as described (*González-Rosa et al., 2011*). Briefly, 6–12 mpf (month post fertilization) fish were anesthetized by immersion into 0.04% tricaine (Sigma, St Louis, MO). A small incision was made through the body wall and the pericardium using microdissection scissors, and a cryoprobe cooled in liquid nitrogen, was placed on the ventricular surface until thawing was observed. Fish were then returned to fresh water for recovery.

## Angiography

Angiography was performed on anesthetized fish by retro-orbital injection (*Pugach et al., 2009*) of Qtracker705 (Invitrogen Q21061MP). Fish were euthanized 2–5 min following injection.

## qRT–PCR

qRT–PCR was carried out as previously described (*Avraham-Davidi et al., 2012*) using the following primers: *vegfc* (*Astin et al., 2014*)

> *cxcl12a*_F 5'- CGTAGTAGTCGCTCTGATGG-3'
> *cxcl12a*_R 5'- TGGGACTGTGTTGACTGTGGAA −3'
> *cxcl12b*_F 5'- GGAGCATCCGAGAGATCAAG-3'
> *cxcl12b*_R 5'- TGTTCTTCAGCTTGGCAATG-3'

## Pharmacological treatments

To reduce heart rate, zebrafish *larvae* were treated daily with 100 µM of the beta-adrenergic antagonist Atenolol (Sigma-Aldrich, A7655) added to the fish water for 2.5 weeks starting at 17 dpf (*Hein et al., 2015*).

To induce cardiomegaly, 8 wpf fish were treated for 3 weeks with 2.5 µg/ml phenylhydrazine hydrochloride (PHZ, Sigma, 78690). To acclimate the fish to PHZ, the first treatment was of 30 min in 1.25 µg/ml PHZ solution. Every other day thereafter, fish were incubated for 1 hr in 2.5 µg/ml PHZ solution followed by 30 min wash in fish water (*Sun et al., 2009*).

To induce Vegfaa-OE, 67 dpf *Tg(βactin2:loxP-mTagBFP-STOP-loxP-vegfaa)*$^{pd262}$ fish were treated for 24 hr with 5 µM 4-hydroxytamoxifen (Sigma, H7904) or 0.05% ethanol (vehicle control).

## Immunohistochemistry and imaging

Zebrafish hearts were isolated from terminally anesthetized zebrafish and fixed briefly (30 s) in 4% PFA/PBS. For immunostaining, hearts were fixed in 4%PFA/PBS overnight, incubated in blocking solution (1% goat serum, 2% Triton X-100, 1%BSA in PBS) for 2–3 days and then with anti-PROX1 antibody (Abcam, 11941, 1:700) in staining solution (1% goat or donkey serum, 0.25% Triton X-100 in PBS) for 4–5 days at 4°C. Following PBST (PBS with 0.25% Triton X-100) washes for 6 hr, hearts were incubated with Alexa Fluor 488 conjugates secondary antibodies (Jackson ImmunoResearch, 111-485-045) diluted in blocking solution for 3–4 days at 4°C and washed again. Hearts were then mounted into glass capillaries (Brand) in 1.5% low-melting point agarose/PBS solution (ROTH) and

imaged using a light sheet Z.1 microscope (Zeiss Ltd.) equipped with 2 sCMOS cameras PCO- Edge, 10X/0.2 excitation objectives and W-Plan Apochromat 20x/0.1 detection (water immersion). In vivo confocal imaging of *larvae* up to 4 wpf was performed using a Zeiss LSM 700 upright confocal microscope (Carl Zeiss) with a W-Plan Apochromat 20 × objective, NA 1.0.

Mouse embryos from timed pregnancies (morning of plug designated E0.5) were fixed in 4% paraformaldehyde (PFA) for 1 hr. Fixed tissues were left intact or sectioned. Immunofluorescence staining was performed in either 1.5 ml tubes with constant rotation (whole mount) or on microscope slides (tissue sections). Primary antibodies in blocking solution (5% goat or donkey serum, 0.5% Triton X-100 in PBS) were incubated overnight at 4°C followed by PBT (PBS with 0.5% Triton X-100) washes for 6 hr. Secondary antibodies diluted in blocking solution were incubated overnight at 4°C and washed again.

Antibodies: VE-Cadherin (BD Pharmingen, 550548; 1:100); PROX1 (R and D Systems, AF2727; 1:300); ERG (Abcam, ab92513, 1:1000); LYVE-1 (eBiosciences, 14-0443-80, 1:100). Secondary antibodies were Alexa Fluor conjugates (488, 555, 594, 633, 635, 647, Life Technologies; 1:250).

Samples were imaged in Vectashield (Vector Labs) using either a Zeiss LSM-700 or Axioimager A2 epifluorescence microscope.

Acid Fuchsin Orange-G (AFOG) staining was performed on paraffin-embedded tissue sections using an AFOG staining kit (DIAPATH, 010307) following manufacturer's instructions. Samples were imaged using Panoramic SCAN II (3DHISTECH) slide scanner.

## Statistical analyses

Coronary blood and lymphatic vessel coverage area and total length, as well as ventricle and OFT size, were calculated from max. projected confocal images of the hearts, using Angiotool (*Zudaire et al., 2011*). OFT lymphatic sprouts and loops were manually counted using ImageJ/Fiji. Color coded local thickness maps of *cxcr4* mutant hearts were generated using ImageJ/Fiji. Scar area following cryoinjury was measured in single section images using ImageJ, and the percent scar area was calculated with respect to the ventricle total area.

Data was analyzed using the unpaired two-tailed Student's t-test assuming unequal variance from at least three independent experiments, unless stated otherwise. Numerical data represent mean ± s.e.m., unless stated otherwise. For the Atenolol experiment ordered logistic regression test was performed.

## Acknowledgements

The authors thank R Hofi, G Almog, N Stettner and A Harmelin (WIS) for excellent animal care. We thank B Weinstein for the *Tg(mrc1a:EGFP)*<sup>y251</sup>, N Mercader for the *Tg(myl7:GFP)*, A Siekmann for the *cxcr4*<sup>um20</sup> and *cxcl12b*<sup>um100</sup>, and E Tzahor for critical reading of the manuscript. Light sheet Imaging was carried out at the 'de Picciotto-Lesser Cell Observatory in memory of Wolf and Ruth Lesser'.

The authors are grateful to all members of the Yaniv laboratory for discussion, technical assistance and continuous support. This works was supported by European Research Council (335605) and (818858) to KY, Binational Science Foundation (2015289) to KY, Minerva Foundation (712610) to KY, the H and M Kimmel Inst. for Stem Cell Research, the Estate of Emile Mimran (SABRA program). KDP is supported by grants from NIH (R01 HL081674, R01 HL131319, and R01 136182), American Heart Association, and the Leducq Foundation. KR is supported by the NIH (RO1-HL128503) and is a New York Stem Cell Foundation - Robertson Investigator. DYRS is supported by funds from the Max Planck Society, the Leducq Foundation and the DFG (project number 394046768) SFB1366/project A4.

## Additional information

### Competing interests

Didier YR Stainier: Senior editor, *eLife*. The other authors declare that no competing interests exist.

## Funding

| Funder | Grant reference number | Author |
|---|---|---|
| H2020 European Research Council | 335605 | Karina Yaniv |
| H2020 European Research Council | 818858 | Karina Yaniv |
| United States-Israel Binational Science Foundation | 2015289 | Karina Yaniv |
| Minerva Foundation | 712610 | Karina Yaniv |
| H and M Kimmel Institute for Stem Cell Research, the Estate of Emile Mimran | SABRA program | Karina Yaniv |
| National Institutes of Health | R01 HL081674 | Kenneth D Poss |
| National Institutes of Health | R01 HL131319 | Kenneth D Poss |
| National Institutes of Health | R01 136182 | Kenneth D Poss |
| American Heart Association | | Kenneth D Poss |
| Fondation Leducq | | Kenneth D Poss Didier YR Stainier |
| National Institutes of Health | RO1-HL128503 | Kristy Red-Horse |
| New York Stem Cell Foundation | Robertson Investigator | Kristy Red-Horse |
| Max-Planck-Gesellschaft | | Didier YR Stainier |
| Deutsche Forschungsgemeinschaft | 394046768 – SFB 1366 | Rubén Marín-Juez Didier YR Stainier |

The funders had no role in study design, data collection and interpretation, or the decision to submit the work for publication.

## Author contributions

Dana Gancz, Conceptualization, Data curation, Formal analysis, Investigation, Visualization, Writing—original draft, Writing—review and editing, Conducted zebrafish experiments; Brian C Raftrey, Formal analysis, Investigation, Visualization, Writing—review and editing, Conducted experiments with mouse hearts; Gal Perlmoter, Formal analysis, Investigation, Visualization, Writing—review and editing, Conducted zebrafish experiments; Rubén Marín-Juez, Investigation, Visualization, Writing—review and editing, Conducted zebrafish experiments; Jonathan Semo, Investigation, Conducted zebrafish experiments; Ryota L Matsuoka, Generated and validated the vegfdbns257 mutants; Ravi Karra, Generated and provided transgenic lines; Hila Raviv, Conducted zebrafish experiments; Noga Moshe, Project administration, Managed fish work; Yoseph Addadi, Methodology; Ofra Golani, Software; Kenneth D Poss, Supervision; Kristy Red-Horse, Visualization, Writing—review and editing, Supervised mouse experiments; Didier YR Stainier, Writing—review and editing, Supervised part of the zebrafish experiments; Karina Yaniv, Conceptualization, Supervision, Funding acquisition, Visualization, Writing—review and editing

## Author ORCIDs

Gal Perlmoter (ID) https://orcid.org/0000-0003-2017-3008
Rubén Marín-Juez (ID) https://orcid.org/0000-0001-5903-7463
Ryota L Matsuoka (ID) https://orcid.org/0000-0001-6214-2889
Ofra Golani (ID) http://orcid.org/0000-0002-9793-236X
Didier YR Stainier (ID) https://orcid.org/0000-0002-0382-0026
Karina Yaniv (ID) https://orcid.org/0000-0001-5638-7150

## Ethics

Animal experimentation: This study was performed in strict accordance with the recommendations in the Guide for the Care and Use of Laboratory Animals of the National Institutes of Health. All of

the animals were handled according to approved institutional animal care and use committee (IACUC) protocols (#01470218-2) of the Weizmann Institute of Science. The protocol was approved by the Committee on the Ethics of Animal Experiments of the Weizmann Institute of Science. All surgery in fish was performed under tricaine anesthesia, and every effort was made to minimize suffering.

### Decision letter and Author response
Decision letter https://doi.org/10.7554/eLife.44153.sa1
Author response https://doi.org/10.7554/eLife.44153.sa2

## Additional files
### Supplementary files
• Transparent reporting form

### Data availability
All data generated or analyzed during this study are included in the manuscript and supporting files.

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
