## [Decision Letter]

**Acceptance summary:**

The study by Gancz et al. investigates the cardiac lymphatic vasculature in the zebrafish. The authors describe in staged images of transgenic lines that the lymphatics develop from the out flow tract (or bulbous arteriosis) progressively in juvenile stages. They show that the OFT lymphatics originate earlier from the ventral facial lymphatics. Several mutants for known regulators of lymphangiogenesis are presented that form abnormal OFT lymphatics in juvenile stages. They then suggest a proposed second origin for cardiac lymphatics in a population of Prox1 (transgene) positive cells that are isolated over the heart blood vasculature and are proposed to later integrate into the growing coronary lymphatics. In mouse, isolated Prox1+ cells of venous origin are identified and lineage traced to a venous origin. Finally, they describe a possible role for lymphatics following injury to the heart. The data provide interesting evidence for different origins of cardiac lymphatics in development and injury. This is a particularly topical question and certainly of interest to the cardiac and vascular fields.

**Decision letter after peer review:**

[Editors’ note: the authors were asked to provide a plan for revisions before the editors issued a binding decision. What follows is the editors’ letter requesting such plan.]

Thank you for sending your article entitled "Distinct origins and molecular mechanisms regulating lymphatic formation during cardiac growth and repair" for peer review at *eLife*. Your article is being evaluated by three peer reviewers, and the evaluation is being overseen by Marianne Bronner as the Senior and Reviewing Editor.

Given the list of essential revisions, including new experiments, the editors and reviewers invite you to respond within the next two weeks with an action plan and timetable for the completion of the additional work. We plan to share your responses with the reviewers and then issue a binding recommendation.

In particular, the reviewers found the mouse work very preliminary and suggest omitting it. It would be important to refine experimentation and convincingly demonstrate the nature and contribution of the isolated clusters. They suggest that you demonstrate the origins of lymphatics in the cryo-injury model, further characterize markers, and provide insight into their potential function. There is also the need for accurate and consistent staging of animals throughout as well as careful controls for the four mutants to ensure phenotypic specificity.

We hope you find the full reviews, attached below, helpful.

Reviewer #1:

In this study Gancz and co-workers make use of different zebrafish reporter and mutant lines to document the existence of lymphatics in the zebrafish heart and to investigate mechanisms underlying cardiac lymphatic growth during heart development and following adult (cryo-) injury. Specifically, the authors used confocal and light-sheet microscopy imaging to characterise lymphatic vessel formation and expansion in the outflow tract (OFT) and ventricular wall of hearts derived from juvenile to adult compound *fli1, flt1_9a, kdrl flt4, lyve1b, prox1a, mrc1, myl7, kfl2a^bns11^, kfl2b^bns12^, cxcr4^um20^, cxcl12b^mu100^*, v*egfc^um18^* or *vegfd^bns257^* double reporter/mutant transgenic fish. This approach led to the identification of two distinct cardiac lymphatic populations in the zebrafish ventricle, which responded differently to pro-lymphangiogenic cues (e.g. VegfC) and which arise separately by sprouting lymphangiogenesis (from the OFT lymphatics) or coalescence of isolated lymphatic clusters located towards the apex of the ventricle. The latter cardiac lymphatic sub-population in particular was found to represent the main source for lymphatic growth following cardiac (cryo-)injury in zebrafish. Lastly, the authors used immunostaining against standard endothelial markers (VE-Cadherin, Prox1), different lineage-tracing models and confocal microscopy imaging to determine the existence of equivalent cardiac lymphatic populations in the developing mouse heart. Isolated Prox1-positive lymphatic clusters were present in the apical region of the developing ventricular wall (both dorsal and ventral sides) until post-natal (P) day 23, a time-point when these clusters became fully integrated in the cardiac lymphatic network. In addition, the authors traced back a modest contribution from the cardinal vein to the isolated lymphatic clusters, excluded contributions from existing coronary vessels/endocardium and suggested a requirement for an additional, yet unidentified, cell source.

In recent years the interest in cardiac lymphatics has increased significantly, particularly after findings demonstrating heterogeneity within the cardiac lymphatic vasculature in mouse and roles for lymphangiogenesis in cardiac repair following myocardial injury, which has important therapeutic implications. There is no report published thus far on the cardiac lymphatics in zebrafish, either during development of following heart injury, and there remains an important need to fully understand the cellular and molecular mechanism(s) governing cardiac lymphatic growth as such this study timely, novel and of widespread interest. Having said that, there are a number of substantial issues with the manuscript in its current form which need addressing:

1) Overall, the findings reported here come across as being somewhat preliminary. The authors aimed to characterise cardiac lymphatic growth in zebrafish and mouse using an array of transgenic models (including reporter, fate-mapping, mutants) at different developmental timepoints, but this is reported in an inconsistent manner with different reporters being used at different developmental timepoints and then compared together, or only one lymphatic marker, rather than binary combinations (at least), being used at each timepoint. This is particularly evident in the characterisation of cardiac lymphatics in zebrafish (Figure 1), where the authors concluded that the isolated ventricular lymphatic clusters only expressed prox1a (Figures 1F and 4A), but do not show equivalent images (same magnification and heart orientation) of hearts from *prox1a/flt4, prox1a/mrc1, lyve1b/prox1a, lyve1b/mrc1* or *prox1a/lyve1b* (e.g. pending availability of a *lyve1b:GFP* or *prox1a:GFP* line) double transgenic fish at a single developmental stage (e.g. 16wpf as in Figure 1F). Also, for some of the mutant lines the authors characterised OFT lymphatics (Figure 2) at 10wpf (*kfl2a* and *kfl2b*), others at 20wpf (*vegfc* and *vegfd*) and others no specific detail of developmental stage (*cxcr4a* and *cxcl12*; also, no images of control OFTs are provided in Figure 2R, S) and no information on controls are provided in the corresponding text or figure legend. The analyses of ventricular lymphatics at 16wpf (Figure 3) appears problematic: given that OFT lymphatic formation (3-4wpf) takes places before lymphatics sprout into the ventricle (12-16wpf), why do the authors investigate OFT lymphatics at 20wpf and ventricular lymphatics at 16wpf in *vegfc* and *vegfd* mutants? To exclude potential developmental delays in the lymphatic growth/expansion the authors need to carry out their time-course studies more consistently across the different mutants.

2) The evidence for isolated lymphatic clusters in the ventricular wall (Figure 4) is not compelling. These are first described as being located towards the apex, but images in Figures 4B (12wpf) and 4E, G (20wpf) highlight prox1a-expressing LECs in the vicinity of the OFT region; how do the authors distinguish between sprouts from OFT lymphatics and putative isolated clusters if only the prox1a reporter line is used here? This is relevant for the *cxcr4a* (Figure 4E) and *vegfc* (Figure 4G) mutant analysis: in Figure 3 the authors showed reduced OFT lymphatics, but how can the authors exclude the possibility that the LECs observed at 20wpf are in fact delayed sprouts from OFT lymphatics? Also, in Figure 4D the authors included a graph detailing numbers of isolated LEC clusters from 8wpf to 16wpf but did not discuss this in the manuscript, nor did they include representative images for these developmental stages. The inclusion of such images is important to support the author's conclusions and make the findings more robust.

3) An important and outstanding question is what is the origin of the isolated LEC clusters in zebrafish? This may be somewhat beyond scope, but is an important omission which would significantly strengthen the overall findings.

4) There is an imbalance between the characterisation of zebrafish and mouse cardiac lymphatics, with the mouse data being descriptive (without functional targeting) and at a more preliminary stage. Arguably the mouse data could be removed without significantly detracting from the manuscript; or if it is to be retained the analyses need to be more in-depth and rigorous. As above, the authors should include (at least) binary combinations of lymphatic markers in their characterisation of cardiac lymphatics to establish the presence of isolated LEC clusters, in particular combine nuclear (Prox1) with membrane (VEGFR3, LYVE1, PDPN, NRP2) staining patterns. Also, images such as shown in Figure 5 would be more informative if channels were shown separately to properly assess whether LECs are isolated from the sprouting plexus and their localisation. An important outstanding question is when do the clusters first appear in the developing mouse heart? This requires a more detailed time- course of assessment.

5) In the mouse the main result arising from the different lineage-tracing approaches employed appears to be that isolated LECs are from a venous source, derived from the cardinal vein or sinus venosus. How do the authors explain the existence of isolated lymphatic clusters that share the same origin as the main lymphatics sprouts? This is a key question that relates to point 3) above for the zebrafish studies, in that the origin of these clusters across species still remains questionable.

6) With regards to the differences between the ApjCreERT2 and Tie2-Cre drivers in labelling murine cardiac lymphatics, the authors need to test the former at earlier time-points to ensure that the venous endothelium which will give rise to LECs is being targeted in an efficient manner. For instance, tamoxifen should be administered at E8.5 plus E9.5 to determine labelling of the jugular lymphatic sac followed by subsequent tracing into the cardiac lymphatics.

7) The isolated LEC clusters apparently become fully integrated within the lymphatic network by P23; what is their functional role in the postnatal heart (intact and following injury)? The authors should at least speculate on this in the Discussion.

8) In the cryo-injury zebrafish studies (Figure 6) the assessment of a role for these isolated lymphatic clusters in repair it is difficult to interpret in the absence of a fate-mapping model, since it is not possible to determine whether the same cell clusters observed during development are being analysed. Also, the authors need to include sham-operated controls and, as above, use the different binary combinations of reporter lines in a consistent manner across the different time-points analysed. In Figure 6G, the authors reported *flt4*-expressing vascular structures, which are negative for prox1a, sprouting into the injured area; could these be blood vessels? Flt4 expression is shared between blood and lymphatic endothelium during development, becoming restricted to the latter at later stages of development, but *flt4* is also activated in neoangiogenic sprouts. This is supported by the observation of vascular structures positive for *kdrl* and *flt4* in Figure 6J.

9) The regions highlighted in Figure 6G-K appear to be in close proximity to the main lymphatic sprouts, rather than isolated in apical areas as is suggested by the authors- this requires further clarification: a combination of membrane and nuclear LEC staining would enable a clear assessment of the clusters and would serve to identify if they are indeed isolated LECs or just leading cells from the sprouting lymphatic plexus.

Reviewer #2:

The study by Gancz et al. investigates the cardiac lymphatic vasculature in the zebrafish. The authors describe in staged images of transgenic lines that the lymphatics develop from the out flow tract (or bulbous arteriosis) progressively in juvenile stages. They show that the OFT lymphatics originate earlier from the ventral facial lymphatics. Several mutants for known regulators of lymphangiogenesis are presented that form abnormal OFT lymphatics in juvenile stages. They then suggest a proposed second origin for cardiac lymphatics in a population of Prox1 (transgene) positive cells that are isolated over the heart blood vasculature and are proposed to later integrate into the growing coronary lymphatics. In mouse, isolated Prox1+ cells of venous origin are identified and lineage traced to a venous origin. Finally, evidence is provided that suggests a possible role for lymphatics following injury to the heart.

The main interest is the proposal for different origins of cardiac lymphatics in development and injury. This is a particularly topical question and certainly of interest to the cardiac and vascular fields. The data provided are suggestive and interesting. However, there are a number of gaps and major concerns throughout the study. Specifically, from many key data points there are important controls missing. Also there are points where interpretation is not fully supported by the current data. These points need to be addressed before further considering publication of this study.

1) Zebrafish develop in a highly variable manner during the larval to juvenile transition. This variation occurs between and within clutches and is very well documented, thus staging with days post fertilisation once animals are beyond larval stages creates inaccuracies (see Parichy et al., 2009). As such, the standard approach to stage juvenile zebrafish is using body length – the authors should see the definitive study and guidelines in Parichy et al., 2009 (especially findings in Figures 3 and 4).

The use of days post fertilisation in this study rather than body length creates concerns because for many comparisons animals may not be of the same developmental age. For example: are the mutants in Figure 2 (*vegfc, vegfd, cxcr4a, cxcl12*) all the same length or are some mutants developmentally delayed? Likewise for the PHZ treated animals. If they are not truly stage matched, the OFT lymphatic phenotypes may just be secondary to developmental delay.

Can the authors please provide accurate staging using length measurements? Minimally this needs to be provided for key observations that could be significantly skewed by natural variation in fish populations at juvenile stages (such as for the mutant and drug treatments).

2) In Figure 2, the authors show that the OFT lymphatics derive from the ventral facial lymphatics. For all mutants analysed at the level of the OFT, the phenotypes could be due to (secondary to) reduced ventral facial lymphangiogenesis rather than suggesting a specific role for these genes in OFT lymphangiogenesis. Especially suggestive is the fact that *vegfc* heterozygotes already have systemically reduced lymphangiogenesis at 5 dpf (Villefranc et al., 2013). Please provide careful quantification and images of the ventral facial lymphatics in these mutants preceding the formation of the OFT lymphatics. Alternatively, temporally inducible transgenic models could be used to show that there are ongoing roles for these genes in OFT lymphangiogenesis.

3) In the mutant comparisons, mutants should be compared with siblings from the same clutch because there can be variation between clutches. This does not appear to be the case for *vegfc* and *vegfd* mutants, which are compared to just one population termed "wt". Please ensure comparisons and statistics are comparing mutants with clutch matched siblings.

4) The prox1a:RFP transgene uses a Gal4, UAS cassette to enhance signal. This system is known to lead to non-specific expression in tissues in some contexts. It has been previously noted that this transgene does not always accurately reflect endogenous Prox1 protein levels (see Koltowska et al., 2015 Figure 2 and Supplementary Figure 1).

In Figure 4, the proposed "clusters" do not express any lymphatic markers other than this transgene. The images appear (to this reviewers eye) to show blood vasculature that is co-expressing RFP rather than genuine distinct clusters of positively identified lymphatic endothelial cells. One interpretation of the current data from zebrafish could be that this expression is non-specific transgene expression in blood vessels over the heart.

Please show more comprehensive analysis of expression of multiple blood vascular and lymphatic markers in these clusters. Do they contain blood? Please also validate the cluster expression of Prox1 with antibody staining.

5) The use of exclusively nuclear markers in Figure 5 in defining the isolated Prox1 positive LEC clusters in mouse is problematic. Some of these "isolated" nuclei are not that far apart (e.g., in Figure 5B and C). Please provide a much more thorough marker analysis of these clusters in mice with more lymphatic and blood vascular markers and including reliable LEC markers that are membranous such as Nrp2, Lyve1 or Podoplanin. The concern is that cells could be connected by long, cytoplasmic/membranous connections not seen with nuclear stains. Further marker and anatomical analyses would significantly clarify the nature of these clusters.

6) In Figure 6, the authors see lymphatics arising in a wound site following cryoinjury using fixed stage analysis. The origins of these lymphatics have not been lineage traced or time-lapse imaged to allow any claims as to their cellular origins. The wound site in Figure 6B for example is adjacent to the major ventricular lymphatic and only one side of the heart is imaged so there could be major lymphatics in the non-imaged half.

These cells could have arisen by detaching from and migrating from the major OFT derived ventricular lymphatics or could have come from other sources based on the analysis provided.

Further data are required to support the authors claims that "… cardiac injury induces the de-novo formation of lymphatics, through a process reminiscent of lymph-vasculogenesis rather than sprouting lymphangiogenesis."

Reviewer #3:

This study by Gancz and colleagues assesses the mechanisms regulating cardiac lymphatic development in zebrafish and mouse. Using several zebrafish transgenic lines, the authors identified two distinct lymphatic populations expressing different transgenic markers and with different origins. One population of lymphatic vessels appears in the OFT at 3-4wpf, expand to cover the entire OFT and finally sprouts towards the ventricle. The authors found that OFT lymphatic vessels originate from a pre-existing "ventral facial lymphatic" vessel that invades the OFT during metamorphosis. The other population of lymphatic vessels derives from coalescence of isolated LECs clusters of unknown origin in the ventricle.

Through qPCR, the authors observed the upregulation of the pro-lymphangiogenic factors *klf2a, vegfc, cxcl12a* and *cxcl12b* in the OFT during metamorphosis, coinciding with the establishment of the cardiac lymphatic system. Genetic mutants were used to investigate the potential role of these factors for this process. Because of embryonic lethality, the authors had to perform their experiments with heterozygous mutants. They found that the distinct lymphatic subsets differentially respond to these pro-lymphangiogenic signals during growth.

The ventricular lymphatics deriving from OFT vessels sprouts in close association with the coronary vasculature. The requirement of the coronary vasculature for lymphatic sprouting was assessed through analysis of *cxcr4a* mutants and PHZ-treated animals.

Similar LEC population was identified in the mouse heart. The origin of these cardiac lymphatics system was assessed using cell lineage tracing experiment, showing that the different subsets do not derives from the local coronary vasculature, nor from the endocardium.

Finally, the regeneration of lymphatic system was assessed in zebrafish using the cryoinjury model. The authors observed differences in the response of the various lymphatic populations to the injury. Prox1a+ vessels rapidly invade the injury area. However, the majority of lymphatic vessels in the regenerating area form de-novo as isolated clusters, suggesting lymph-vasculogenesis rather than sprouting lymphaniogenesis as the main mechanism for lymphatics regeneration.

Overall, this is an interesting study of high standards. The findings contribute to our knowledge about development of cardiac lymphatics in zebrafish. However, the conclusions based on analysis of mutants are sometimes not convincingly supported by the results. Furthermore, the part with the cryoinjury model contains descriptive data, without testing the role of lymphatics in cardiac regeneration.

- "we detected a significant reduction in OFT lymphatic coverage (Figure 2J-M), with under-developed vessels displaying fewer sprouts that those of their wt siblings". In the Figure 2M, the graph indicates that the percentage of lymphatic coverage was higher in the mutants than in wt, which is in contradiction with the statement of the authors.

- ".., we could not observe any defects in OFT lymphatics of *vegfd* mutant fish (Figure 2N, P, Q)…" The *vegfd* mutant was generated for this study using the Crispr/Cas9 system. Do these mutants display any phenotype, indicative of a gene loss of function?

- "Our results suggest that while VegfC is a key determinant of OFT lymphatic formation, the Cxcr4/Cxcl12 axis is mostly involved in patterning and remodelling of the OFT lymphatic plexus". I don't see any differences between the phenotype of *vegfc* and *cxcr4a* mutants (Figure 2O and R). I found that this conclusion is too strong for these data. How can we conclude that VegfC is a key determinant of OFT lymphatic formation, given that lymphatic vessels were still formed? The authors should describe more precisely the phenotype of the mutants and prove that the lymphatic function is also altered.

It would be also interesting to test if cardiac lymphatic vessel development in the ventricle is also impaired in these mutants.

- ".… suggesting a possible role for the major coronary vessels in guiding lymphatic sprouts". The authors cannot exclude that the treatment itself influences directly lymphatic development.

[Editors' note: Formal revisions were requested, following approval of the authors’ plan of action. Further revisions were then requested prior to acceptance, as described below.]

Thank you for submitting your article "Distinct origins and molecular mechanisms regulate lymphatic formation during cardiac growth and repair" for consideration by *eLife*. Your article has been reviewed by three peer reviewers, and the evaluation has been overseen by Marianne Bronner as the Senior and Reviewing Editor. The reviewers have opted to remain anonymous.

The reviewers have discussed the reviews with one another and the Editor has drafted this decision to help you prepare a revised submission.

Summary:

The major findings, describing the development and response to injury of the zebrafish cardiac lymphatics are of sufficient interest for publication in *eLife* and it is noted this is to be accompanied by the "sister" paper by Harrison and colleagues, also under revision. While the paper is much improved, the reviewers feel that a few changes are still needed as summarized below. The full reviews also are included below for further clarification.

Essential revisions:

i) Additional work is required to definitively prove the existence of LEC clusters in zebrafish hearts as distinct from the main venous-derived lymphatic plexus; specific experiments using multiple stainings and z-stack confocal projections, etc are suggested in the report.

ii) The mouse studies remain quite preliminary and still require additional work to add any insight over and above what has already been published. The clusters are not clearly demonstrated and there is no insight into origin. At the moment it detracts from the fish studies, so the mouse data should be dropped and instead the authors should focus on definitively demonstrating that LEC clusters exist in the fish heart.

Reviewer #1:

In this revised manuscript, Gancz and co-workers have improved the original submission with new data and convincingly document the development of lymphatic vessels in the zebrafish heart after the larva-to-juvenile transition, as well as their response/requirement following cryoinjury in the adult heart. The new data include the binary combination of markers to characterise cardiac lymphatics (including antibody staining to complement their transgenic lines), the inclusion of images from age/size-matched controls, removal of data deemed too preliminary (e.g. *kfl2a/b* data) and inclusion of new models, such as *flt4*^-/-^ and vegfAa-OE. Having said that, some of the issues noted in the original submission still remain to be addressed. In particular, the authors still need to more convincingly demonstrate the following:

1) The existence of isolated lymphatic clusters in the developing zebrafish heart and in the adult zebrafish heart following cryo-injury.

The observation that the prox1a:RFP transgene is also active in blood endothelium (subsection “Morphological and molecular heterogeneity of zebrafish cardiac lymphatics”, last paragraph) makes it challenging to determine whether the putative clusters reported here are truly lymphatic in nature, or represent an artefact of the transgenic reporter (Figure 4 and Figure 1—figure supplement 2). The authors employed immunostaining using an antibody against prox1 to validate their transgenic approach, and whilst this revealed that the OFT and connecting ventricular lymphatic branches were clearly positive for nuclear expression of prox1 the majority of RFP-labelled ventricular vessels were negative leaving a question mark as to the source of the prox1a:RFP+ cells (Figure 1—figure supplement 2D versus E). Moreover, a similar concern rests with the other two reporters employed (*flt4BAC:mCitrine, lyve1b:dsRed2*), in that both *flt4* and *lyve1* have been shown to be expressed in blood endothelium. Consequently, how can the authors be sure that the putative ventricular lymphatic clusters reported are not subsets of blood endothelial cells activating the mCitrine and dsRed reporters, respectively? Equally, how do these transgenic lines "behave" in the cryoinjury setting, i.e. does injury (hypoxia, cell death?) induce ectopic expression of the transgenes? These are critical issues confounding the interpretation of the data generated through the use of the different transgenic reporters and need to be fully addressed to support the notion of isolated LEC clusters. In this regard, the authors should carefully map out the overlap in blood and lymphatic endothelium, both in development and post-injury, for each of the reporters in combination with specific antibodies against lymphatic markers, akin to *prox1* in Figure 4 and Figure 1—figure supplement 2.

2) A requirement for coronary vessels in cardiac lymphatic expansion in the ventricular wall.

The new data derived from the vegfAa-OE model in a *vegfC*+/- genetic background does not support the authors' conclusion for a requirement for coronary vessels in ventricular lymphatic vessel expansion. In this model *vegfC* haploinsufficiency is ubiquitous, rather than specific to coronary endothelium, and other cell types have been reported to express *vegfC* in the developing heart, such as epicardial cells (Chen et al., 2014). Therefore, the reduction of lymphatic growth is quite likely to be downstream of a general reduction in levels of *vegfC* arising from multiple cell types. Likewise, the observation of increase lymphatic expansion following vegfAa-OE may reflect an overall increase in ventricle size and the need for an adequate lymphatic coverage, akin to PHZ treatment. The authors need to discuss these possibilities and tone down their conclusions.

3) The existence of isolated lymphatic clusters in the developing mouse heart.

Unfortunately the mouse data still remain preliminary and inconclusive regarding the presence of discrete clusters of LECs of non-venous origin. The authors have included a new panel c in Figure 5, but this is suboptimal as the region shown is a high-power view and, therefore, the relationship to the main lymphatic network developing in the E15.5 murine heart is unclear. The authors should provide a low-power view containing the entire ventricular surface. Also, in order to confirm that the "multicellular structures" marked with the blue arrowheads are detached from the unmarked vascular structures, the authors need to include a representative z-stack. Specifically, re Figure 5E the z-stack depth in the lower power image is different from the high-power inset, and the region of interest appears to be connected to the main lymphatic network as suggested by the continuous VE-Cadherin labelling in the inset. It is evident that VE-Cadherin labels lymphatic endothelium at later stages of heart development (compare Figure 5A-D with Figure 5E) and as such the authors need to highlight this in the manuscript. Statements such as "the Prox1+ cells gradually expanded between days E15.5-E17.5 to form multicellular structures (Figure 5D, E)" need to be revised and toned down, as the authors are not lineage-tracing these clusters (nor their progeny). Finally, the inclusion of the Ccbe mutant data here does not add to the overall novelty of the findings and is limited to the mouse studies. Taken together, the main issues raised for this part of the study still remain and if the authors are intent on including these mouse studies (not required in the opinion of this reviewer) then they need to revise further to go beyond simply confirming previous reports in mouse, describing non-venous sources contributing to cardiac lymphatics (lineage-tracing data in Figure 5).

Specific points:

- In Figures 1, 3, 4, 6 and Figure 1—figure supplement 2, the inset boxes representing high-power views of regions of interest are covering the low-power/whole heart views and need to be separated into distinct panels.

- To confirm lack of connections of putative clusters to proximal ventricular lymphatics, the authors need to provide a representative confocal z-stack.

- Why is the coverage of *prox1a:RFP* labelled vessels in the ventricle so extensive, when compared to *lyve1b:dsRed* or *flt4:mCitrine* reporters? This links to point 1) above.

- For the cryoinjury model, why is it that all the sham images exhibited reduced or almost no ventricular lymphatic vessels, compared to injured heart at similar age/stage (Figure 6—figure supplement 1)? Are these images truly representative of the stages at which cryoinjury took place?

- Whilst in revision, a new study from the Porrello lab has been published reporting that Vegfc/d-dependent regulation of the lymphatic vasculature during cardiac regeneration is influenced by injury context (Vivien et al., 2019); the authors should discuss this work in the revised manuscript.

Reviewer #2:

The authors have provided a much-improved manuscript in response to reviewer questions. The additional transgenic markers, antibody stains and angiograms have clarified cell identity and improved the description of cardiac lymphangiogenesis. The mouse lineage tracing experiments are improved with further markers. The genetic studies using VegfA-Oe and Vegfc mutants in zebrafish, and observations that isolated clusters and major lymphatic tracts respond differently in different mutants, improve the value of this study.

---

## [Author Response]

Reviewer #1:[…] There are a number of substantial issues with the manuscript in its current form which need addressing:1) Overall, the findings reported here come across as being somewhat preliminary. The authors aimed to characterise cardiac lymphatic growth in zebrafish and mouse using an array of transgenic models (including reporter, fate-mapping, mutants) at different developmental timepoints, but this is reported in an inconsistent manner with different reporters being used at different developmental timepoints and then compared together, or only one lymphatic marker, rather than binary combinations (at least), being used at each timepoint. This is particularly evident in the characterisation of cardiac lymphatics in zebrafish (Figure 1), where the authors concluded that the isolated ventricular lymphatic clusters only expressed prox1a (Figures 1F and 4A), but do not show equivalent images (same magnification and heart orientation) of hearts from prox1a/flt4, prox1a/mrc1, lyve1b/prox1a, lyve1b/mrc1 or prox1a/lyve1b (e.g. pending availability of a lyve1b:GFP or prox1a:GFP line) double transgenic fish at a single developmental stage (e.g. 16wpf as in Figure 1F).

We fully agree with the reviewer on that the presentation of our data was not “clean” enough in the original version of our manuscript. We have added images of combined lymphatic transgenes at comparable stages (based on age and body length), using similar magnifications and heart orientations for all lymphatic subsets, and have made sure that the data are presented this way throughout the entire manuscript.

Also, for some of the mutant lines the authors characterised OFT lymphatics (Figure 2) at 10wpf (kfl2a and kfl2b), others at 20wpf (vegfc and vegfd) and others no specific detail of developmental stage (cxcr4a and cxcl12; also, no images of control OFTs are provided in Figure 2R, S) and no information on controls are provided in the corresponding text or figure legend.

We apologize for the inconsistencies in the presentation of our data. We now provide additional images and information for all mutants: All OFT lymphatic analyses were performed on 16-20 wpf/16-24mm fish. WT siblings or age-matched control images were added as well. These data are included in new Figure 2.

The analyses of ventricular lymphatics at 16wpf (Figure 3) appears problematic: given that OFT lymphatic formation (3-4wpf) takes places before lymphatics sprout into the ventricle (12-16wpf), why do the authors investigate OFT lymphatics at 20wpf and ventricular lymphatics at 16wpf in vegfc and vegfd mutants? To exclude potential developmental delays in the lymphatic growth/expansion the authors need to carry out their time-course studies more consistently across the different mutants.

The variability among the analysed stages derives primarily from the fact that some of these fish were compared based on body length, in order to exclude potential developmental delays resulting from size vs. age differences. To avoid potential misunderstanding, we restricted all the OFT lymphatic analyses to 16-24mm fish (WTs and mutants) and ventricular lymphatic analyses to 25-30mm fish (WTs and Mutants). All figures and quantitations have been updated accordingly and explanations about measurement procedures have been added to the Materials and methods section.

2) The evidence for isolated lymphatic clusters in the ventricular wall (Figure 4) is not compelling. These are first described as being located towards the apex, but images in Figures 4B (12wpf) and 4E, G (20wpf) highlight prox1a-expressing LECs in the vicinity of the OFT region; how do the authors distinguish between sprouts from OFT lymphatics and putative isolated clusters if only the prox1a reporter line is used here? This is relevant for the cxcr4a (Figure 4E) and vegfc (Figure 4G) mutant analysis: in Figure 3 the authors showed reduced OFT lymphatics, but how can the authors exclude the possibility that the LECs observed at 20wpf are in fact delayed sprouts from OFT lymphatics?

We thank the reviewer for this comment. The initial distinction between OFT/ventricular lymphatic sprouts and isolated LECs, in both wt and mutant animals, was based on whether they are connected or not to a pre-existing lymphatic vessel. While ventricular lymphatics remain connected to their OFT lymphatic source, isolated clusters are not connected to any vessel. To further confirm these results, we now provide several images of the isolated clusters, labeled by different marker combinations, and throughout different stages, which demonstrate that these clusters are not connected to other lymphatic vessels. The data can be found in Figures 1F, H and Figure 4.

We have also carried out a thorough characterization of the numbers of isolated LECs in *cxcr4, vegfc* and *flt4* mutants, as well as following VegfAa over-expression and PHZ treatment. We found that unlike ventricular lymphatics, isolated lymphatic clusters are normally present in *cxcr4* mutants, and are not affected by vegfaa over-expression and PHZ treatment. In contrast, they are absent from *flt4* mutant hearts and are markedly reduced in *vegfc* mutants, emphasizing the requirement of Flt4/VegfC signaling for development of all cardiac lymphatic subsets. These data are included in new Figure 4.

Also, in Figure 4D the authors included a graph detailing numbers of isolated LEC clusters from 8wpf to 16wpf but did not discuss this in the manuscript, nor did they include representative images for these developmental stages. The inclusion of such images is important to support the author's conclusions and make the findings more robust.

We have added to Figure 4 representative images for the analysed stages as well as quantification of isolated clusters in double LEC transgenic lines.

3) An important and outstanding question is what is the origin of the isolated LEC clusters in zebrafish? This may be somewhat beyond scope, but is an important omission which would significantly strengthen the overall findings.

We agree with the reviewer about the importance of identifying the origins of the isolated clusters, but concur also with the appreciation that these experiments fall beyond the scope of this manuscript, especially given the range of the revision period in *eLife*. While we have attempted to analyse the putative contribution of additional cell types, we find the data too preliminary and not conclusive to be added to this manuscript.

4) There is an imbalance between the characterisation of zebrafish and mouse cardiac lymphatics, with the mouse data being descriptive (without functional targeting) and at a more preliminary stage. Arguably the mouse data could be removed without significantly detracting from the manuscript; or if it is to be retained the analyses need to be more in-depth and rigorous. As above, the authors should include (at least) binary combinations of lymphatic markers in their characterisation of cardiac lymphatics to establish the presence of isolated LEC clusters, in particular combine nuclear (Prox1) with membrane (VEGFR3, LYVE1, PDPN, NRP2) staining patterns. Also, images such as shown in Figure 5 would be more informative if channels were shown separately to properly assess whether LECs are isolated from the sprouting plexus and their localisation. An important outstanding question is when do the clusters first appear in the developing mouse heart? This requires a more detailed time- course of assessment.

We believe that the identification of isolated lymphatic clusters in mammals in addition to fish, is an important finding which contributes to the understanding of cardiac lymphatic formation across species. As shown in Figure 5, the clusters first appear at E14.5, and continue to expand between days E15.5-E17.5, forming multicellular lumenized structures. By day P23 the isolated lymphatic clusters are no longer detected. Binary combination of nuclear (Prox1) and membrane (LYVE1) were added to Figure 5, which include also channel separation to better demonstrate the lack of connections between isolated lymphatic clusters and the main lymphatic vessels. In addition, we now demonstrate that the formation of both “regular” cardiac lymphatics and isolated lymphatic clusters is VEGFR3/VegfC-dependent, and therefore abolished in *Ccbe* KO mice, thus supporting the data acquired in fish. These data are included in new Figure 5.

5) In the mouse the main result arising from the different lineage-tracing approaches employed appears to be that isolated LECs are from a venous source, derived from the cardinal vein or sinus venosus. How do the authors explain the existence of isolated lymphatic clusters that share the same origin as the main lymphatics sprouts? This is a key question that relates to point 3) above for the zebrafish studies, in that the origin of these clusters across species still remains questionable.

Our lineage tracing analyses in mouse indeed revealed that both isolated clusters and main lymphatic sprouts are partially derived from a venous origin. One explanation for these results could be that these two lymphatic populations originate from different veins (such as the cardinal vein and sinus venosus). Alternatively, they could arise in the same vein but employ different mechanisms of migration and sprouting to reach the heart, thus ensuring proper and perhaps faster lymphatic coverage of the heart. Finally, it could also be possible that some cells detach from the parent lymphatic vessel and migrate through, as has been shown for LECs in the lung, which bud from extra-pulmonary lymphatics and migrate as single cells or small clusters into the developing lung (Kulkarni et al., 2011). We will add these points to the Discussion.

Notably, the relative low fraction of LECs labeled by ApjCreERT2 (20-30%), compared to the high recombination efficiency of this Cre driver (~90%), hints at putative additional non-venous sources with major contribution to both isolated and “traditional” cardiac lymphatics. Yet, at this stage, we have not been able to definitely prove the alternative origins of these cells

6) With regards to the differences between the ApjCreERT2 and Tie2-Cre drivers in labelling murine cardiac lymphatics, the authors need to test the former at earlier time-points to ensure that the venous endothelium which will give rise to LECs is being targeted in an efficient manner. For instance, tamoxifen should be administered at E8.5 plus E9.5 to determine labelling of the jugular lymphatic sac followed by subsequent tracing into the cardiac lymphatics.

We agree with the reviewer that ensuring efficient targeting of the venous endothelium giving rise to LECs, namely the CCV, by early activation of ApjCreERT2 (E8.5+e9.5) is an important control. We lineage traced Apj CreER expressing cells using the mTmG reporter. Tamoxifen was dosed to the mother at e9.5 and e10.5, and embryos were collected at e11.5. Using this dosing strategy, we saw efficient labeling of the cardinal vein, and labeling in the jugular lymph sac. Some of the mTmG+, Prox1+ cells appear to be budding off the cardinal vein. These results are consistent with the conclusion that the ApjCreER lineage traced cardiac lymphatics could be derived from the cardinal vein and lymph sacs. These results were added to Figure 5—figure supplement 1.

7) The isolated LEC clusters apparently become fully integrated within the lymphatic network by P23; what is their functional role in the postnatal heart (intact and following injury)? The authors should at least speculate on this in the Discussion.

This is indeed an interesting question. Unlike zebrafish, in which the cardiac lymphatic system continues growing through adulthood, mouse cardiac lymphatics are fully formed by P23 and the isolated LEC clusters are fully integrated by then. Yet, the fact that at least part of them appear to arise from different sources, may hint at different specializations during pathological conditions. A paragraph discussing these possibilities has been added to the text.

8) In the cryo-injury zebrafish studies (Figure 6) the assessment of a role for these isolated lymphatic clusters in repair it is difficult to interpret in the absence of a fate-mapping model, since it is not possible to determine whether the same cell clusters observed during development are being analysed. Also, the authors need to include sham-operated controls and, as above, use the different binary combinations of reporter lines in a consistent manner across the different time-points analysed.

We agree with the reviewer about the importance of identifying the origins of the isolated clusters following cardiac injury. We have performed additional experiments and could not find LECs at the injury site co-expressing the myeloid marker Lys or the retinoic acid synthesizing enzyme raldh2, known to be present in the epicardium and endocardium following injury. The images are provided in Author response image 1 for the reviewers’ convenience, but we have opted to omit them from the manuscript. We included a paragraph in the Discussion listing potential origins for LECs following injury including recruitment of isolated LECs. In addition, sham-operated animals and binary combinations of reporter lines across the different time points were added to Figure 6—figure supplement 1 as requested.

**Author response image 1. respfig1:** LECs in the injured area do not co-express endocardial, epicardial or myeloid markers. (**a**) Section of a *Tg(flt4*:mCitrine) heart at 7 dpi. Cardiomyocytes are immunostained with anti-MHC antibody (red). Epicardium and injury activated endocardium are immunostained with anti-retinoic acid (RA)-synthesizing enzyme Raldh2 (white). White dotted lines delineate the injured area. Insets show high-magnification of yellow dashed box. *Flt4* positive cells in the injured area are not labeled with Raldh2 (**b**) *Tg(flt4:*mCitrine*);Tg(lyz:*dsRed) heart at 6dpi. White dotted lines delineate the injured area. Insets show high-magnification of yellow dashed box. *Flt4* positive cells in the injured area do not express the myeloid marker *lys*. Scale bars are 200µm.

In Figure 6G, the authors reported flt4-expressing vascular structures, which are negative for prox1a, sprouting into the injured area; could these be blood vessels? Flt4 expression is shared between blood and lymphatic endothelium during development, becoming restricted to the latter at later stages of development, but flt4 is also activated in neoangiogenic sprouts. This is supported by the observation of vascular structures positive for kdrl and flt4 in Figure 6J.

We agree with the reviewer that some of the *flt4* positive sprouts may represent blood endothelial cells. while, previous publications (He et al., 2017; Marín-Juez et al., 2016) demonstrated that neovascularization following cardiac injury involves only sprouting from pre-existing blood vessels, a new study now demonstrates collateral assembly via migration and coalescence of isolated arterial cells following MI in mice (Das et al., 2019). While both *prox1a* and *flt4* transgenes may individually label blood as well as lymphatic ECs, we found that cells expressing both transgenes are of lymphatic identity and are not labeled by intravascular injection of Qdot705 (Figure 1—figure supplement 2C). We therefore used *prox1a/flt4* (Figure 6F, I, J) as well as *prox1a/mrc1a* (Figure 6H) transgenic combinations to confirm the lymphatic identity of isolated LECs in the injured area.

9) The regions highlighted in Figure 6G-K appear to be in close proximity to the main lymphatic sprouts, rather than isolated in apical areas as is suggested by the authors- this requires further clarification: a combination of membrane and nuclear LEC staining would enable a clear assessment of the clusters and would serve to identify if they are indeed isolated LECs or just leading cells from the sprouting lymphatic plexus.

We thank the reviewer for this suggestion. We complemented our data with additional views of the injured hearts, clearly demonstrating the lack of connections with other vessels. These were added to the new Figure 6—figure supplement 1A, B, D, F.

Reviewer #2:[…] There are a number of gaps and major concerns throughout the study. Specifically, from many key data points there are important controls missing. Also there are points where interpretation is not fully supported by the current data. These points need to be addressed before further considering publication of this study.1) Zebrafish develop in a highly variable manner during the larval to juvenile transition. This variation occurs between and within clutches and is very well documented, thus staging with days post fertilisation once animals are beyond larval stages creates inaccuracies (see Parichy et al., 2009). As such, the standard approach to stage juvenile zebrafish is using body length – the authors should see the definitive study and guidelines in Parichy et al., 2009 (especially findings in Figures 3 and 4).The use of days post fertilisation in this study rather than body length creates concerns because for many comparisons animals may not be of the same developmental age. For example: are the mutants in Figure 2 (vegfc, vegfd, cxcr4a, cxcl12) all the same length or are some mutants developmentally delayed? Likewise for the PHZ treated animals. If they are not truly stage matched, the OFT lymphatic phenotypes may just be secondary to developmental delay.Can the authors please provide accurate staging using length measurements? Minimally this needs to be provided for key observations that could be significantly skewed by natural variation in fish populations at juvenile stages (such as for the mutant and drug treatments).

We fully agree with the reviewer’s comment and apologize for not having added this information in our original submission. In this revised version in addition to age, we provide standard body length measurements (the distance from the snout to the caudal peduncle) for all the experiments presented, including those comparing wt vs. mutants and control vs. treatments. Importantly, throughout all our experiments, special care was taken to include only fish of same age *and size* to avoid bias resulting from potential developmental differences.

2) In Figure 2, the authors show that the OFT lymphatics derive from the ventral facial lymphatics. For all mutants analysed at the level of the OFT, the phenotypes could be due to (secondary to) reduced ventral facial lymphangiogenesis rather than suggesting a specific role for these genes in OFT lymphangiogenesis. Especially suggestive is the fact that vegfc heterozygotes already have systemically reduced lymphangiogenesis at 5 dpf (Villefranc et al., 2013). Please provide careful quantification and images of the ventral facial lymphatics in these mutants preceding the formation of the OFT lymphatics. Alternatively, temporally inducible transgenic models could be used to show that there are ongoing roles for these genes in OFT lymphangiogenesis.

We appreciate the reviewer’s comment. We have documented the development of the ventral facial lymphatic vessel in all mutants (*vegfc, vegfd, cxcr4, cxcl12b* and *flt4*) and have detected no defects in its formation. These data are included in Figure 2—figure supplement 2.

3) In the mutant comparisons, mutants should be compared with siblings from the same clutch because there can be variation between clutches. This does not appear to be the case for vegfc and vegfd mutants, which are compared to just one population termed "wt". Please ensure comparisons and statistics are comparing mutants with clutch matched siblings.

Throughout all experiments included in our revised version, mutants are compared with clutch matched siblings or aged/size-matched WTs in the case of certain homozygous adult mutants (e.g. *vegfd* and *cxcl12b*).

4) The prox1a:RFP transgene uses a Gal4, UAS cassette to enhance signal. This system is known to lead to non-specific expression in tissues in some contexts. It has been previously noted that this transgene does not always accurately reflect endogenous Prox1 protein levels (see Koltowska et al., 2015 Figure 2 and Supplementary Figure 1).In Figure 4, the proposed "clusters" do not express any lymphatic markers other than this transgene. The images appear (to this reviewers eye) to show blood vasculature that is co-expressing RFP rather than genuine distinct clusters of positively identified lymphatic endothelial cells. One interpretation of the current data from zebrafish could be that this expression is non-specific transgene expression in blood vessels over the heart.Please show more comprehensive analysis of expression of multiple blood vascular and lymphatic markers in these clusters. Do they contain blood? Please also validate the cluster expression of Prox1 with antibody staining.

We appreciate the reviewer’s comment. In this revised version of our manuscript we provide a thorough characterization of the *prox1a* expressing vessels and assess the specificity of the transgene. As pointed out by the reviewer, we detected sporadic expression of the *prox1a* transgene in few arterial capillaries, that were co-labelled by the arterial enhancer *Tg(flt1_9a_cFos:*GFP*)^wz2^* and highlighted by intravascular injection of Qdot705 (Figure 1—figure supplement 2B). In contrast, *prox1a/flt4* double labeled vessels (in *prox1a;flt4* double transgenic zebrafish) were devoid of intravascular injected Qdots, confirming their lymphatic identity. Hence, we identify two vessel populations labelled by the *prox1a* transgene- one, where its expression fully overlapped with that of the *flt4* reporter and was devoid of Qdot705 labelling (Figure 1—figure supplement 2C),and a second one labelled only by *prox1a* and Qdot705 (Figure 1—figure supplement 2C).We further analysed the isolated lymphatic clusters during development (Figure 4) and following cardiac injury (Figure 6) using this transgenic combination. We injected Qdots intravascularly and verified that neither the ventricular lymphatics nor the isolated clusters co-labeled by *prox1a* and additional transgene are connected to the blood circulation (see new Figure 1—figure supplement 2D, E and Figure 4F).

In order to investigate whether both populations indeed express Prox1, we carried out immunostaining with anti-Prox1 antibody. As seen in Figure 1—figure supplement 2only*prox1a* positive LECs, but not *prox1a* positive blood ECs were labelled by the Prox1 antibody (Figure 1—figure supplement 2E, insets), suggesting that the expression in blood ECs represents an artefact of the transgenic reporter.

5) The use of exclusively nuclear markers in Figure 5 in defining the isolated Prox1 positive LEC clusters in mouse is problematic. Some of these "isolated" nuclei are not that far apart (e.g., in Figure 5B and C). Please provide a much more thorough marker analysis of these clusters in mice with more lymphatic and blood vascular markers and including reliable LEC markers that are membranous such as Nrp2, Lyve1 or Podoplanin. The concern is that cells could be connected by long, cytoplasmic/membranous connections not seen with nuclear stains. Further marker and anatomical analyses would significantly clarify the nature of these clusters.

Binary combination of nuclear (Prox1) and membrane (LYVE1) staining were added to Figure 5. These clearly demonstrate the lack of connections between isolated lymphatic clusters and the main lymphatic sprouts.

6) In Figure 6, the authors see lymphatics arising in a wound site following cryoinjury using fixed stage analysis. The origins of these lymphatics have not been lineage traced or time-lapse imaged to allow any claims as to their cellular origins. The wound site in Figure 6B for example is adjacent to the major ventricular lymphatic and only one side of the heart is imaged so there could be major lymphatics in the non-imaged half.

We agree with the reviewer about our inability to define the cellular origins of the isolated clusters, and this is clearly mentioned throughout the manuscript. However, there are certain facts that we have carefully documented and quantified, for which we can make clear statements. For instance, while the wound site may sometimes be adjacent to major ventricular lymphatics, we carefully verified that the isolated lymphatic clusters at the injury site are not directly connected to any pre-existing lymphatic vessel. Our conclusions are based on imaging of the heart from multiple angles using lightsheet microscopy, which confirm that the isolated LECs are not connected to pre-existing ventricular lymphatics. The images showing different views of injured hearts have been added to Figure 6D, F, H, I and Figure 6—figure supplement 1A, B, D, F.

These cells could have arisen by detaching from and migrating from the major OFT derived ventricular lymphatics or could have come from other sources based on the analysis provided.Further data are required to support the authors claims that "… cardiac injury induces the de-novo formation of lymphatics, through a process reminiscent of lymph-vasculogenesis rather than sprouting lymphangiogenesis."

We believe that the data presented in Figure 6D, F, H, I and Figure 6—figure supplement 1A, B, D, F demonstrate that LECs in the injury site, regardless of their origin, are not connected to other lymphatic vessels, and initially appear as isolated clusters. Therefore, we concluded that their appearance is “reminiscent of lymph-vasculogenesis”. As stated above, we have included additional views of the injured hearts in new Figure 6—figure supplement 1A, B, D, to further support this claim.

We agree with the reviewer that without live imaging or defined lineage-tracing we cannot claim anything about their origins (this caveat appears in the Discussion), yet we believe our data strongly supports the fact that these vascular structures do not form through lymphangiogenesis.

Reviewer #3:[…] Overall, this is an interesting study of high standards. The findings contribute to our knowledge about development of cardiac lymphatics in zebrafish. However, the conclusions based on analysis of mutants are sometimes not convincingly supported by the results. Furthermore, the part with the cryoinjury model contains descriptive data, without testing the role of lymphatics in cardiac regeneration.- "we detected a significant reduction in OFT lymphatic coverage (Figure 2J-M), with under-developed vessels displaying fewer sprouts that those of their wt siblings". In the Figure 2M, the graph indicates that the percentage of lymphatic coverage was higher in the mutants than in wt, which is in contradiction with the statement of the authors.

We thank the reviewer for pointing out this mistake. The sentence should state, "we detected a significant *increase* in OFT lymphatic coverage”. However, since our new data revealed major roles for the Vegfc/Flt4 and Cxcl12/Cxcr4 pathways in OFT lymphatic development, we decided to omit the *klf2* data that was inconclusive and perhaps less relevant to this process. If however, the reviewers think otherwise, we will add them back to the manuscript.

- ".., we could not observe any defects in OFT lymphatics of vegfd mutant fish (Figure 2N, P, Q)…" The vegfd mutant was generated for this study using the Crispr/Cas9 system. Do these mutants display any phenotype, indicative of a gene loss of function?

We appreciate the reviewer’s comment. *vegfd^bns257^* homozygous mutants indeed exhibit compromised facial lymphatic development, consistent with previous reports, however no major defects were observed in the VFL, from which OFT lymphatics sprout. The position and morphology of these vessels, which run in parallel to the ventral aorta was similar in wt and mutant fish. We now provide additional data describing the generation and early phenotypic characterization of these mutants in Figure 2—figure supplement 3.

- "Our results suggest that while VegfC is a key determinant of OFT lymphatic formation, the Cxcr4/Cxcl12 axis is mostly involved in patterning and remodelling of the OFT lymphatic plexus". I don't see any differences between the phenotype of vegfc and cxcr4a mutants (Figure 2O and R). I found that this conclusion is too strong for these data. How can we conclude that VegfC is a key determinant of OFT lymphatic formation, given that lymphatic vessels were still formed? The authors should describe more precisely the phenotype of the mutants and prove that the lymphatic function is also altered.

In the revised version of our manuscript we show that while *vegfc^+/-^* animals display significantly reduced OFT lymphatic coverage, *cxcr4* mutants show defective patterning of OFT lymphatics, with no overall changes in lymphatic coverage. Since *vegfc^+/-^* animals still carry a wt copy of the *vegfc* gene (homozygous mutants do not survive through adulthood), which could lead to a partial phenotype, we analysed also OFT lymphatic development in homozygous adult *flt4* mutants. As seen in new Figure 2I-K, OFT lymphatics are nearly absent in *flt4* mutants, strongly supporting and expanding our original findings.

It would be also interesting to test if cardiac lymphatic vessel development in the ventricle is also impaired in these mutants.

Impaired ventricular lymphatic development was also observed in *vegfc^+/-^* and *cxcr4* mutants. In addition, our new data reveal that ventricular lymphatics are completely absent in *flt4* mutants.

These results were added to new Figure 3.

- ".… suggesting a possible role for the major coronary vessels in guiding lymphatic sprouts". The authors cannot exclude that the treatment itself influences directly lymphatic development.

Following the reviewer comment, we carried out additional experiments to further verify the involvement of coronary vessels in guiding lymphatic vessels:

1) In addition to PHZ treatment, we induced myocardial expansion and increased blood vessel coverage by over-expressing the angiogenic factor VegfAa (Karra et al., 2018). Our new results demonstrate that lymphatic coverage, as well as the diameter of ventricular lymphatics, is significantly increased in these hearts (new Figure 3E-G).

2) In order to ascertain whether VegfAa overexpression enhances lymphangiogenesis indirectly (through increased angiogenesis) or by direct action of VegfAa on LECs, we over-expressed VegfAa in *vegfc^+/-^* heterozygous animals and assessed whether despite reduced VegfC production, there is still lymphatic over-growth, which could have pointed to a direct effect of VegfAa on LECs. As seen in new Figure 3Q-T and Figure 3—figure supplement 1J, while depletion of VegfC had no effect on VegfAa induced cardiomegaly and hyper-vascularization, ventricular lymphatic growth was abolished, suggesting that enhanced coronary formation induces ventricular lymphatic growth in a VegfC-dependent manner.

[Editors' note: further revisions were requested prior to acceptance, as described below.]Essential revisions:i) Additional work is required to definitively prove the existence of LEC clusters in zebrafish hearts as distinct from the main venous-derived lymphatic plexus; specific experiments using multiple stainings and z-stack confocal projections, etc. are suggested in the report.

Our manuscript provides solid evidence for the existence of isolated LEC clusters in zebrafish. We base this statement in the following lines of evidence:

1) We provide images of different transgene combinations, at different stages, which clearly demonstrate that the isolated LEC clusters are not connected to neither blood nor lymphatic vessels (Figure 1F, H and Figure 4). This is in contrast to the main ventricular lymphatics that remain connected to their OFT lymphatic source.

2) In the previous assessment of our manuscript, certain concerns were raised regarding the specificity of the *prox1a* transgene in labeling lymphatic vessels. In order to address these comments, we carried out Prox1 immunostaining experiments in combination with intravascular injection of Qdot705. Comparison between the *prox1* transgenic reporter and Prox1 immunostaining, revealed the presence of two different vessel populations labelled by the *prox1a* transgene: (i) Lymphatic ECs, where the expression of the *prox1a* transgene fully overlaps with that of the *flt4* reporter and is devoid of Qdot705 labelling (Figure 1—figure supplement 2C), and (ii) a population of arterial capillaries, co-labelled by the arterial enhancer *Tg(flt1_9a_cFos:*GFP*)^wz2^* (Figure 1—figure supplement 2B), which is connected to the blood circulation as demonstrated by Qdot705 labelling (Figure 1—figure supplement 2C). Based on these results, we decided to use this combinatorial matrix to assess the nature of the isolated LECs: intravascular injection of Qdot705 into double transgenic lines (*flt4BAC:mCitrine;prox1a:*RFP) confirmed that neither the ventricular lymphatics nor the isolated clusters are connected to the blood circulation (Figure 1—figure supplement 2D, E and Figure 4F). It is important to note that all analyses of the isolated LEC clusters during development (Figure 4) and following cardiac injury (Figure 6), were carried out using double transgenic combinations.

3) Following assessment of our revised version, reviewer #1 raises new concerns about two of the other transgenic reporters used in this study (“a similar concern rests with the other two reporters employed (flt4BAC:mCitrine, *lyve1b:dsRed2*), in that both *flt4* and *lyve1* have been shown to be expressed in blood endothelium”. In this case, we disagree with the reviewer’s comment:

a) Both the *flt4BAC:mCitrine and lyve1b:dsRed2* reporters have been shown to be expressed by both venous and lymphatic endothelial cells **in the developing embryo** (Van Impel et al., 2014; Nicenboim et al., 2015; Okuda et al., 2012; Marin-Juez et al., 2016). The expression of these transgenes in the adult zebrafish heart however, has not been thoroughly characterized. Our results, following intravascular injection of Qdot705 indicate that the expression of these reporters is restricted to lymphatic vessels in the adult zebrafish heart (Figure 1I, K and Figure 4F). Specifically, we detect the presence of perfused blood vessels (Figure 1K, blue only) that are not labelled by the *flt1_9a_cFos:*GFP arterial enhancer (Figure 1K, green +blue) or by the *lyve1b:dsRed2* reporter(Figure 1K, red), which we conclude are venous vessels.Hence, we are confident that the cells labeled by the *flt4BAC:mCitrine* and *lyve1b:dsRed2* reporters in the adult heart are for the most part LECs.

b) To further confirm these results, we analysed cardiac lymphatics using an additional transgenic reporter- *mrc1a:GFP,* whichhas been shown to label LECs in juvenile zebrafish (Jung et al., 2017), and find that its expression fully overlaps with that of the other reporters.

c) Finally, it is important to note that the expression of the lymphatic reporters is first detected over the ventricle at around ~12wpf, when ventricular lymphatics begin sprouting, despite of the fact that a fully functional blood vasculature (veins and arteries) is present much earlier (~8 wpf) (Figure 3 and Harrison et al., 2015).

4) Regarding the reviewer’s request that “the authors should carefully map out the overlap in blood and lymphatic endothelium, both in development and post-injury, for each of the reporters in combination with specific antibodies against lymphatic markers….”

As explained above, we provide extensive evidence for the overlapping expression of the 4 different lymphatic reporters utilized in this study. Moreover, since we noticed that the *prox1* reporter displayed additional blood vessel expression, we carried out Prox1 immunostaining to further confirm our findings. I must emphasize that this was not an easy task- Most available antibodies do not work properly in zebrafish, especially in adult animals, being this one of the main reasons for the wide use of transgenic reporters instead. To the best of our knowledge there is only one additional study describing immunostaining of lymphatic vessels in adult zebrafish (Shimoda and Isogai, Acta Histochem Cytochem. 2012), whose results fully concur with our own. Unfortunately, no antibodies against the zebrafish proteins are available for the other markers. Hence, it will not be feasible to carry out these experiments in the timeframe of this revision.

5)“Equally, how do these transgenic lines "behave" in the cryoinjury setting, i.e. does injury (hypoxia, cell death?) induce ectopic expression of the transgenes?”

We believe that the use of multiple transgenic lines, all labeling the same set of LEC clusters after injury, significantly minimizes the chances that they represent ectopic expression of the reporters. It seems unlikely that 4 different transgenic reporters will render ectopic expression in similar sets of cells, all resembling lymphatic vessels, but we cannot completely rule this out. While the behavior of these lines in settings of hypoxia and/or cell-death might by itself be interesting, we believe this characterization falls beyond the scope of this manuscript.

6) “To confirm lack of connections of putative clusters to proximal ventricular lymphatics, the authors need to provide a representative confocal z-stack’)”.

We provide full confocal z-stacks for the following images: (Figure 1F, H; Figure 4B, G; Figure 6G, F, H; Figure 6—figure supplement 1H; Figure 7Q) showing lack of connections between the isolated LEC clusters and the ventricular lymphatics. Yet, given the curved architecture of the heart, we believe that single plane images are more informative than the multiple z-stack confocal planes. Such image of a single z-stack is provided in Figure 4E inset.

7) Finally, we have also carried out a thorough characterization of the isolated LECs in *cxcr4, vegfc* and *flt4* mutants, as well as following VegfAa over-expression and PHZ treatment. We found that unlike ventricular lymphatics, isolated lymphatic clusters are normally present in *cxcr4* mutants, and are not affected by VegfAa over-expression and PHZ treatment. In contrast, they are absent from *flt4* mutant hearts and are markedly reduced in *vegfc*^+/-^ animals (Figure 4). These results confirm that the isolated LECs represent a separate lymphatic population, which responds differently than ventricular lymphatics/blood vessels to the same signaling cues.

Overall, we believe that the thorough characterization of the OFT, ventricular and isolated LECs presented in our manuscript, which includes analysis of different combinations of all available lymphatic transgenic reporters (*lyve1, flt4*, mrc1a and *prox1*), antibody immunostaining, and functional assays (microangiography) to distinguish between blood and lymphatic vessels, provides strong foundation for the existence of a separate population of isolated LECs.

ii) The mouse studies remain quite preliminary and still require additional work to add any insight over and above what has already been published. The clusters are not clearly demonstrated and there is no insight into origin. At the moment it detracts from the fish studies, so the mouse data should be dropped and instead the authors should focus on definitively demonstrating that LEC clusters exist in the fish heart.

We strongly disagree with the reviewer’s comment. The findings describing the existence of a separate population of isolated lymphatic cells in the murine heart have not been previously reported, and as such they are novel.

We agree with the reviewer that we were unable to provide insight into the origins of the lymphatic clusters that appear during cardiac development. However, we have thoroughly analyzed the morphology of the clusters at many different time points during embryonic and postnatal development. Numerous confocal images were taken of over 100 hearts during preliminary investigations and in the data reported here in which we find that developing hearts contain lymphatic clusters. In the spirit of *eLife*, we believe it is important that these findings are published as they will surely open new avenues for future research.

In response to the below comments, please see that we have included additional images that better depict what we have observed many times during our experiments (I. E., Orthogonal sections of our confocal images as well as low and high-power Z stack examples). We will note that although a non-venous origin was shown for cells within the lymphatic sprouts (Klotz et al., 2015), the authors did not describe the presence of lymphatic clusters and no other publication since then has either. We therefore believe that although we could not show origins of the clusters, reporting their presence is an important part of this study as it shows an analogous structure to that thoroughly investigated in zebrafish. We hope that our request to keep these data will be favorably considered.

Reviewer #1:[…] Some of the issues noted in the original submission still remain to be addressed. In particular, the authors still need to more convincingly demonstrate the following:[…] 3) The existence of isolated lymphatic clusters in the developing mouse heart.Unfortunately the mouse data still remain preliminary and inconclusive regarding the presence of discrete clusters of LECs of non-venous origin.

We have added additional panels to Figure 5—figure supplement 1A-C showing that the clusters are indeed discrete. Using Lyve1 to mark lymphatic cell membranes, we found that some lymphatic cells had no connections with any other neighboring lymphatic cell (Figure 5—figure supplement 1A). Furthermore, using VE-cadherin and Prox1 immunofluorescence, we provide a whole ventricle, 20X, and 40X images of a typical cluster with associated orthogonal views and a video through all the Z-stacks, which confirms that it does not communicate with other lymphatic sprouts or coronary vessels (Figure 5—figure supplement 1B and C, Figure 5—video 1). It is important to note that we do not make any claims regarding venous vs. non-venous origins of these cells, but just describe the results from our different lineage-tracing studies, which show mostly negative (but accurate) results.

The authors have included a new panel C in Figure 5, but this is suboptimal as the region shown is a high-power view and, therefore, the relationship to the main lymphatic network developing in the E15.5 murine heart is unclear.

We have added a low magnification image of the heart that shows the main lymphatic vessel so it is now possible to understand the location of the isolated clusters relative to the other vessels (Figure 5—figure supplement 1A).

The authors should provide a low-power view containing the entire ventricular surface.

A low power image of the entire ventricle is now included for both Lyve1 immunofluorescence and VE-cadherin/Prox1 (Figure 5—figure supplement 1A-C, Figure 5—video 1).

Also, in order to confirm that the "multicellular structures" marked with the blue arrowheads are detached from the unmarked vascular structures, the authors need to include a representative z-stack. Specifically, re Figure 5E the z-stack depth in the lower power image is different from the high-power inset, and the region of interest appears to be connected to the main lymphatic network as suggested by the continuous VE-Cadherin labelling in the inset.

We reimaged this heart and have now included images that contain Z stacks through the same depth of tissue (Figure 5—figure supplement 1B and C). We have also highlighted a typical isolated LEC cluster (with orthogonal views and a video through the Z stack, Figure 5—figure supplement 1B and C, Figure 5—video 1) which is not connected to VE-Cadherin+ blood or lymphatic vessels. The clusters are frequently close to coronary vessels, in this case two cell lengths away (Figure 5—figure supplement 1C), but this is likely because coronary vessels are dense within the ventricle.

It is evident that VE-Cadherin labels lymphatic endothelium at later stages of heart development (compare Figure 5A-D with Figure 5E) and as such the authors need to highlight this in the manuscript.

Thank you for this suggestion. We have included this point in the text.

Statements such as "the Prox1+ cells gradually expanded between days E15.5-E17.5 to form multicellular structures (Figure 5D, E)" need to be revised and toned down, as the authors are not lineage-tracing these clusters (nor their progeny).

This statement was removed and instead we describe the clusters as simply being present during development and not observable in three-week-old hearts.

Finally, the inclusion of the Ccbe mutant data here does not add to the overall novelty of the findings and is limited to the mouse studies. Taken together, the main issues raised for this part of the study still remain and if the authors are intent on including these mouse studies (not required in the opinion of this reviewer) then they need to revise further to go beyond simply confirming previous reports in mouse, describing non-venous sources contributing to cardiac lymphatics (lineage-tracing data in Figure 5).

We feel that the manuscript benefits from the Ccbe1 data because it is evidence that, like in Zebrafish, the VEGFc pathway is required for the presence of isolated LEC clusters. Other studies reported that isolated lymphatic clusters of non-venous origins in the mesentery are independent of the VEGFC receptor VEGFR3 (Stanczuk, 2015). Therefore, it is a key piece of information regarding the developmental mechanisms of cardiac LEC clusters.

Specific points:- In Figures 1, 3, 4, 6 and Figure 1—figure supplement 2, the inset boxes representing high-power views of regions of interest are covering the low-power/whole heart views and need to be separated into distinct panels.

We separated all high-power views into distinct panels as requested

- To confirm lack of connections of putative clusters to proximal ventricular lymphatics, the authors need to provide a representative confocal z-stack.

We provide full confocal z-stacks for the following images: (Figure 1F, H; Figure 4B, G; Figure 6G, F, H; Figure 6—figure supplement 1H; Figure 7P, Q) showing lack of connections between the isolated LEC clusters and the ventricular lymphatics. Yet, given the curved architecture of the heart, we believe that single plane images are more informative than the multiple z-stack confocal planes. Such image of a single z-stack is provided in Figure 4E inset.

- Why is the coverage of prox1a:RFP labelled vessels in the ventricle so extensive, when compared to lyve1b:dsRed or flt4:mCitrine reporters? This links to point 1) above.

As explained above, the *prox1a:RFP* transgene labels also a subset of arterial capillaries, highlighted also by the arterial enhancer *flt1-9a:GFP* and therefore its coverage is more abundant than other lymphatic markers. This observation further supports the fact that the other reporters (*flt4, lyve1, mrc1a*) specifically label LECs and not blood ECs.

- For the cryoinjury model, why is it that all the sham images exhibited reduced or almost no ventricular lymphatic vessels, compared to injured heart at similar age/stage (Figure 6—figure supplement 1)? Are these images truly representative of the stages at which cryoinjury took place?

The sham operations were carried out together with the cryo-injury experiments and are indeed representative. We have replaced the image in Figure 6—figure supplement 1C. In addition, a recent paper also reports no differences in cardiac lymphatics following sham operation (Vivien et al., 2019).

- Whilst in revision, a new study from the Porrello lab has been published reporting that Vegfc/d-dependent regulation of the lymphatic vasculature during cardiac regeneration is influenced by injury context (Vivien et al., 2019); the authors should discuss this work in the revised manuscript.

We have added a paragraph to the Discussion referring this study.